# Reactivating hippocampal-mediated memories during reconsolidation to disrupt fear

Stephanie L. Grella [1,2], Amanda H. Fortin[1], Evan Ruesch[1], John H. Bladon [1,3], Leanna F. Reynolds[1], Abby Gross[1], Monika Shpokayte[1], Christine Cincotta[1], Yosif Zaki[4] & Steve Ramirez [1] ✉

Memories are stored in the brain as cellular ensembles activated during learning and reactivated during retrieval. Using the Tet-tag system in mice, we label dorsal dentate gyrus neurons activated by positive, neutral or negative experiences with channelrhodopsin-2. Following fear-conditioning, these cells are artificially reactivated during fear memory recall. Optical stimulation of a competing positive memory is sufficient to update the memory during reconsolidation, thereby reducing conditioned fear acutely and enduringly. Moreover, mice demonstrate operant responding for reactivation of a positive memory, confirming its rewarding properties. These results show that interference from a rewarding experience can counteract negative affective states. While memory-updating, induced by memory reactivation, involves a relatively small set of neurons, we also find that activating a large population of randomly labeled dorsal dentate gyrus neurons is effective in promoting reconsolidation. Importantly, memory-updating is specific to the fear memory. These findings implicate the dorsal dentate gyrus as a potential therapeutic node for modulating memories to suppress fear.

Maladaptive conditioned fear, caused by dysregulated fear circuits, plays a significant role in the etiology of anxiety disorders such as specific phobias and post-traumatic stress disorder (PTSD). PTSD can develop in individuals who have experienced a traumatic event and it is often characterized by persistent memories of the trauma[1]. Consequently, contextual fear-conditioning (CFC), which is highly conserved across species[2], has been used as a representative model in animals to study certain aspects of PTSD, such as fear generalization, exaggerated fear responses, and enhanced stress reactivity[3–5]. The most widely used CFC paradigms involve pairing an emotionally neutral conditioned stimulus (CS) such as a training context, with an aversive unconditioned stimulus (US) like a foot shock that typically elicits activity bursts that lead to conditioned freezing responses in rodents. A learned association emerges, and the CS acquires aversive properties that facilitate retrieval of the conditioned fear memory in the absence of the US. In rodent models, this results in a conditioned fear response

upon re-exposure to the context demonstrating this learned relationship[6]. In humans, pathological conditioned fear can occur for decades even in the absence of the exact context in which the traumatic event took place.

In spite of the fact that anxiety disorders are extremely prevalent in the general population, and many individuals experience pathological anxiety as a form of an exaggerated fear state, there are few ways to attenuate maladaptive conditioned fear. Reconsolidation, however, has potential as a therapeutic mechanism for diminishing Pavlovian fear[7]. Reconsolidation theory posits that memories become destabilized during recall as they enter a transient state of malleability where they can be modulated during the time it takes them to restabilize[7,8]. Despite the long history of experimental reconsolidation-related interventions using a variety of pharmacological agents, behavioral treatments and stimulation protocols to disrupt or enhance memory[9–11], these studies have yielded mixed results. Only recently has

[1]Department of Psychological & Brain Sciences, Boston University, Boston, MA 02215, USA. [2]Department of Psychology, Loyola University Chicago, Chicago, IL 60660, USA. [3]Department of Psychology, Brandeis University, Waltham, MA 02453, USA. [4]Department of Neuroscience, Icahn School of Medicine at Mount Sinai, New York, NY 10029, USA. ✉e-mail: dvsteve@bu.edu

the potential for developing improved reconsolidation-based treatments and novel interventions been recognized[12,13]. Nevertheless, most effective therapies for PTSD are trauma-focused, meaning the treatment focuses on the memory of the traumatic event[14].

Memory is thought to be stored in the sparse activity patterns of neuronal populations within a distributed network[15–17], or as Wilder Penfield described memory as "the writing left behind by conscious experience"[18]. We often refer to these ensembles, active during memory encoding, as memory traces or engrams[19,20], and these engrams are reactivated during retrieval[17,21]. Findings from several studies have shown that specific memories, including fear memories, can be disrupted by inhibition of associated engrams[15,22–26]. Specifically, the dorsal dentate gyrus (dDG) of the hippocampus is important for encoding contextual fear memories[27–30], and has been implicated in the pathophysiology of a number of anxiety disorders[28,31]. Of particular relevance to PTSD, contextual information, which includes more than spatial information, has the capacity to modulate fear and safety[4]. Valence (e.g., negative memories) can be considered an aspect of context, which has the potential to promote exaggerated fear responses and fear generalization through associations formed in the hippocampus. Importantly, the DG also plays a role in disambiguating trauma-related and non-trauma-related contextual information[32,33] as well as in extinction learning[34] and PTSD patients exhibit impairments in both[3]. Moreover, we have previously shown that artificial reactivation of a positive memory stored in the dDG can acutely rescue stress-induced, depression-related behavior[35].

Here, we propose an innovative intervention based on the hypothesis that using optogenetics to artificially reactivate a previously formed, dDG-mediated memory during reconsolidation will permanently alter and disrupt the original fear memory. We used the Tet-tag system[36] to label dDG neurons activated by exposure to positive, neutral or negative experiences with channelrhodopsin-2 (ChR2). Mice were subsequently fear conditioned and given a fear memory recall test wherein these tagged neurons were optically reactivated. We hypothesized that this intervention during the reconsolidation window would update the fear memory with attributes from the competing engram, thereby reducing the behavioral expression of conditioned fear. Moreover, as we have previously shown that stress-induced behaviors can be rescued by optically reactivating dDG cells previously active during a positive experience[35] and others have shown that positive emotions counteract a subset of aftereffects of negative emotions[37], we proposed that this effect would be more pronounced when the competing engram was associated with a positive experience compared to a neutral or negative experience.

Here we show that reconsolidation-based hippocampal interference induced by optical reactivation of a competing, positive memory is sufficient to update a fear memory at the ensemble level resulting in an attenuation of maladaptive conditioned fear.

## Results

### Artificial reactivation of hippocampal-mediated memories during fear memory reconsolidation reduces fear acutely and enduringly

We used a viral, activity-dependent, and inducible neuronal tagging strategy in wild-type c57BL/6 mice (Fig. 1a). Male mice were injected with virus (either ChR2 or eYFP) and implanted with bilateral optic fibers before being taken off DOX to open a tagging window[17,38]. They were split into three groups, and each assigned a differentially-valenced behavioral experience (Fig. 1a). All mice were placed into a novel clean cage and either left undisturbed (neutral)[21], placed with a female (positive)[35,39], or placed into a restraint tube with air holes (negative)[35] and then placed into the cage. Mice were returned to their home cages 1 h later back on DOX to close the tagging window. The following day, mice were fear conditioned (FC) in context A and 24 h later given a 20 min recall test in the same context. During this test, in

which we assessed conditioned fear (i.e., freezing) as a proxy for retrieval of the associative fear memory, we simultaneously stimulated the tagged dDG ensembles during the first (F10) or last half (L10) of the session rather than the entire session, as we did not want to risk heat damage to the brain[40] and wanted to compare light-on and light-off periods in a within-subject manner. We initially hypothesized that reactivating a positive memory during the last half of the recall session would promote reconsolidation since the fear memory would already be online. This would potentially alter the fear memory ensemble, updating it with positive attributes from the experience resulting in decreased freezing at subsequent time points. We aimed to specifically weaken the strength of the CS-US association by dampening the acquired aversion to the CS[41] and to alter the original fear memory through reconsolidation. Therefore, we chose a session length not longer than 20 min to ensure that our optical manipulation would be introduced during the short window post-memory reactivation when reconsolidation occurs[42] and not during extinction learning[43]. This strategy permits us to measure real-time decreases in freezing with optical stimulation during recall. Moreover, it permits us to measure any long-lasting effects of our manipulation, and we thus extended our assessment to include two extinction sessions to test for stress-induced reinstatement after an immediate shock in context B. The shock was delivered in a new context in under 2 s so mice would not form a contextual representation of the environment, and, therefore not form a new associative fear memory, but would still experience stress. This method allowed us to model fear generalization and heightened stress reactivity as an example of maladaptive conditioning since the stressor was delivered in a different context[44–46].

Based on previous studies[39,41,47], mice first were FC using a 4-shock protocol (Fig. 1b) wherein they exhibited freezing in a stepwise manner, increasing with each successive shock presented (Supplementary Fig. 1a–c). We saw this pattern of freezing for all experiments (Supplementary Fig. 1a–h). Mice were returned to the context the next day for a fear memory recall test (Fig. 1c). With L10 stimulation, mice in positive and negative-ChR2 groups demonstrated a real-time reduction in freezing compared to mice in the neutral-ChR2 group and to eYFP controls respectively. While there was a natural decline in freezing across the session due to the absence of shock, these mice showed a significantly steeper decline. While freezing levels generally declined across extinction days, no group differences were observed during extinction (Fig. 1d and Supplementary Fig. 1a) or immediate shock (Fig. 1e). During reinstatement (Fig. 1f), we saw less freezing in negative-ChR2 mice compared to neutral-eYFP mice, and in general, eYFP control mice froze more than experimental ChR2 mice at reinstatement compared to immediate shock (Fig. 1g). In contrast, post-shock freezing following fear-conditioning (Fig. 1h) was reduced in the F10 condition (Fig. 1i), only for positive-ChR2 mice compared to eYFP controls, which occurred specifically in the last 10 min of the session. Here, neutral-ChR2 mice extinguished more quickly; however, no group differences were observed during extinction (Fig. 1j and Supplementary Fig. 1b). As expected, there were no group differences during immediate shock (Fig. 1k). During reinstatement, both positive and neutral-ChR2 groups demonstrated reduced fear compared to eYFP controls, while negative-ChR2 mice did not (Fig. 1l). Again, control mice froze more than experimental mice at reinstatement compared to immediate shock (Fig. 1m), and this was a more pronounced effect. Therefore, we adopted the F10 protocol for all subsequent experiments. However, as an additional control, we added an experiment to assess how effective artificial reactivation of a positive memory during the middle portion of the session (M10) would be compared to F10 or L10 stimulation during recall (Supplementary Fig. 2a). During acquisition of the FC response, mice in all three conditions showed greater freezing post-shock (Supplementary Fig. 2b–g). During recall, the stimulation caused real-time decreases in freezing in the ChR2 group in the F10 and L10 conditions but not in the M10 condition

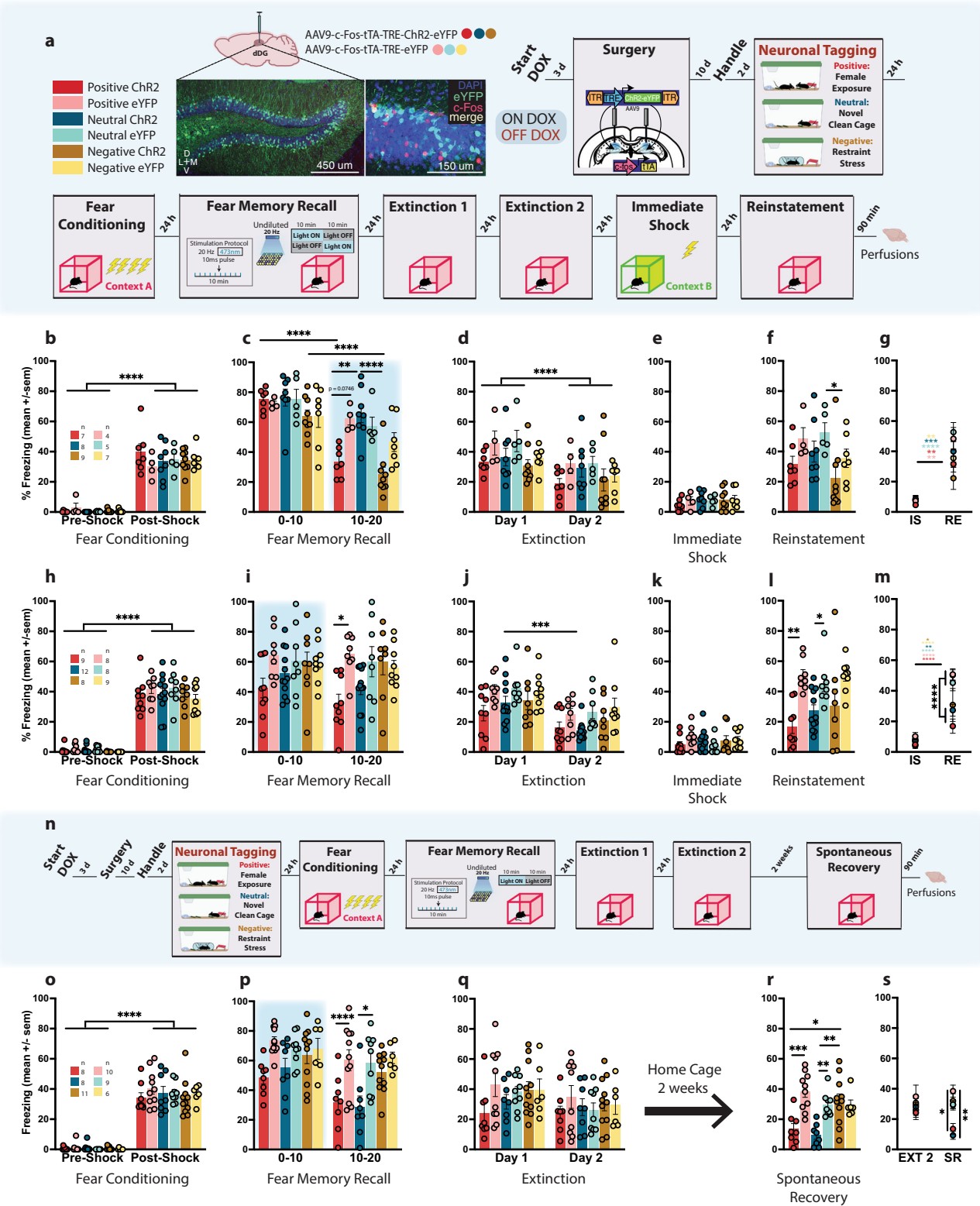

(Supplementary Fig. 2h–j). F10 ChR2 mice continued to show decreased freezing in the latter half of the recall session in the absence of stimulation (Supplementary Fig. 2j). All three stimulation protocols produced decreased freezing in ChR2 mice in the first 3 min of extinction, but group differences were only seen with F10-stimulated mice (Supplementary Fig. 2k–m). Across the first extinction session, fear expression was decreased in F10 or M10 ChR2 groups (Supplementary Fig. 2n–p). There were no differences in immediate shock (Supplementary Fig. 2q–s). Finally, we again saw diminished freezing

for both F10 and M10 ChR2 groups during reinstatement (Supplementary Fig. 2t–y).

Next, we assessed long-term effects of our manipulation. We replicated the above findings with a similar experimental design. However, instead of giving mice a reinstatement test after immediate shock, we left mice undisturbed in their home cage for 2 weeks after extinction and then gave them a test for the spontaneous recovery of fear (Fig. 1n). Twenty-four hours after fear-conditioning (Fig.1o), during recall, both positive and neutral-ChR2 groups froze less in the last half

**Fig. 1 | Artificial reactivation of hippocampal-mediated memories during fear memory reconsolidation reduces fear acutely and enduringly. a** Viral strategy and experimental design. dDG cells encoding positive, neutral, and negatively-valenced behavioral epochs were tagged off-DOX (orange). Mice were FC (context A) and 24 h later given a recall session during which cells previously tagged were artificially reactivated in either the first (F10), or last half (L10) of the session. Across next 2 days, mice were given 2 EXT sessions. The following day, to reinstate fear responding, mice received an IS (context B) and the next day were tested for RE. **b** Mice showed greater freezing post-shock (three-way RM ANOVA: $F(1,33) = 448.3$, $P < 0.0001$). **c** During recall, L10 mice in positive ($P = 0.022$) and negative ($P < 0.0001$) ChR2-groups showed less freezing compared to neutral-ChR2 mice but not eYFP-controls (three-way RM ANOVA: $F(2,34) = 4.665$, $P = 0.0162$, Time × Valence × Virus). Freezing declined faster for both groups ($P < 0.0001$). **d–f** No group differences during EXT, or IS. At RE, negative-ChR2 mice showed less freezing than neutral-eYFP mice ($p = 0.0449$). **g** eYFP controls froze more than experimental mice at RE (three-way RM ANOVA: $F(1,34) = 5.704$, $P = 0.0226$, Virus × Day); $F(1,34) = 3.969$, $P = 0.0282$, Virus × Valence). **h** A separate cohort was FC demonstrating greater freezing post-shock (three-way RM ANOVA: $F(1,48) = 761.4$, $P < 0.0001$). **i** With F10 stimulation, only positive-ChR2 mice showed reduced fear in the last half of the session compared to eYFP controls (three-way RM ANOVA: $F(1,48) = 7.737$; $P = 0.0077$). **j** Neutral-ChR2 mice extinguished most rapidly (three-

way RM ANOVA: $F(1,48) = 57.75$, $P < 0.0001$, Time; $F(1,48) = 10.99$; $P = 0.0017$). **k** No group differences seen during IS. **l** During RE, both positive ($P = 0.0016$) and neutral ($P = 0.0031$) ChR2-groups showed less fear compared to eYFP controls, while negative-ChR2 mice did not (two-way ANOVA: $F(1,48) = 26.97$, $P < 0.0001$). **m** Control mice froze more than experimental mice at RE (three-way RM ANOVA: $F(1,48) = 24.66$, $P < 0.0001$, Day × Virus). **n** A separate cohort was tested on SR. **o** Mice demonstrated greater freezing post-shock (three-way RM ANOVA: $F(1,46) = 685.2$, $P < 0.0001$). **p** Positive ($P = 0.0251$) and neutral ($P = 0.0266$) ChR2 mice showed reduced fear in the last half of recall compared to eYFP controls (three-way RM ANOVA: $F(1,46) = 16.75$, $P = 0.0002$, Time; $F(1,46) = 28.15$, $P < 0.0001$). **q** No group differences observed during EXT. **r** In a test for SR, both positive ($P = 0.0004$, $P = 0.0122$) and neutral ($P = 0.0070$, $P = 0.0011$) mice showed less freezing compared to eYFP controls and compared to negative-ChR2 mice (two-way ANOVA: $F(2,46) = 6.894$, $P = 0.0024$, Valence × Virus). **s** Positive and neutral ChR2 mice continued to exhibit less fear 2 weeks after EXT compared to both positive and neutral-eYFP mice, and both negative groups (three-way RM ANOVA: $F(2,46) = 3.784$, $P = 0.0301$, Valence × Virus; $F(1,46) = 7.859$, $P = 0.0074$, Day). Data represented as means ± s.e.m. *$P < 0.05$, **$P < 0.01$, ***$P < 0.005$, ****$P < 0.00$. dDG dorsal dentate gyrus, DOX doxycycline, EXT extinction, IS immediate shock, RE reinstatement, SR spontaneous recovery. Source data provided as a Source Data file.

---

of the session compared to eYFP controls (Fig. 1p). We saw no group differences during extinction (Fig. 1q and Supplementary Fig. 1c). Consistent with effects seen during recall, the spontaneous recovery test revealed that both positive and neutral-ChR2 mice froze less compared to eYFP controls and compared to the negative-Chr2 group demonstrating that our manipulation produced enduring effects on fear memory retrieval processes evident two weeks after extinction (Fig. 1r, s).

Importantly, we showed the decreases in freezing observed after tagging and stimulating a positive memory, cannot be attributed solely to the viral injection nor light stimulation alone. Specifically, we found that reactivation of a tagged positive memory (female exposure) (Supplementary Fig. 3a), after mice were FC to show increased post-shock freezing (Supplementary Fig. 3b, c), resulted in less freezing in only ChR2 mice that received laser stimulation. This was in comparison to ChR2 mice did not receive stimulation, eYFP mice that did receive stimulation, and to the no virus/no laser group. This was true during recall (Supplementary Fig. 3d), on the first day of extinction, and during the first 3 min of extinction (Supplementary Fig. 3e, f). As expected, we saw no group differences during immediate shock (Supplementary Fig. 3g) but saw a significant decrease in freezing in the ChR2-Laser On group compared to all other groups at reinstatement (Supplementary Fig. 3h, i). Moreover, this effect was only present when mice received artificial reactivation of a tagged a memory in the FC context (Supplementary Fig. 4a). Next, we tagged a positive memory (female exposure) and then FC mice that demonstrated post-shock freezing (Supplementary Fig. 4b). When this memory was reactivated in a novel context and mice were returned to the conditioning context 24 h later, freezing did not decrease in the ChR2 group during recall (Supplementary Fig. 4c). Freezing remained similar to eYFP controls throughout extinction and reinstatement (Supplementary Fig. 4d, e). These findings provide evidence that our manipulation occurs via reconsolidation, as it is specific to when mice experience natural recall of the fear memory, which in this case, is prompted by exposure to the conditioning context.

### Valence matters: Artificial reactivation of a neutral home cage experience during reconsolidation is not sufficient to disrupt a fear memory

The above results illustrate that hippocampal interference resulting from reactivation of positive or neutral engrams is more effective at reducing conditioned fear than engrams associated with a negative experience. To further gauge the importance of valence, first, we tested if the novel clean cage experience was indeed "neutral", rather than

positive or negative given that novel stimuli can engage a complex set of approach and avoidance dopaminergic pathways related to salience, reward, and neophobia[48–52]. Therefore, for the neutral component of the next experiment, we used a home cage[38] experience instead, where mice were left undisturbed. Secondly, we asked whether a separate positive experience that did not involve female exposure was sufficient to reduce fear. Consequently, we used acute cocaine exposure[53] as our next positive experience. And finally, we asked if the inability to reduce freezing via stimulation of a negative engram during fear memory recall was the result of those memories overlapping. To test this, we tagged dDG cells active when mice were FC in context C as our next negative experience as this interfering engram would theoretically be composed of some of the same cells as the fear memory acquired in context A due to generalization.

To address these questions, we again opened a tagging window off DOX and labelled a positive, neutral, or negative memory in the dDG (Fig. 2a). Mice assigned to negative groups were initially FC in context C demonstrating significant post-shock freezing (Fig. 2b). The following day they were FC as before in context A. Mice FC the previous day showed higher freezing than the other groups pre- and post-shock (Fig. 2c). During recall, negative ChR2 mice continued to exhibit more freezing compared to other ChR2 groups (Fig. 2d), which was observed throughout the session. However, there were no real-time decreases in freezing in any of the ChR2-groups, compared to eYFP counterparts in any part of the recall session (Fig. 2d). No group differences were observed during extinction (Fig. 2e and Supplementary Fig 1d) nor immediate shock (Fig. 2f). Optical stimulation of the home cage memory was not sufficient to compete with the fear memory given that during reinstatement, we observed that only positive-ChR2 mice showed less freezing compared to eYFP controls and the negative groups (Fig. 2g). Negative-ChR2 mice demonstrated equal freezing to controls (Fig. 2g, h). These results corroborate our previous findings showing that optical stimulation of a competing positive memory, but not a neutral or negative memory, is sufficient to disrupt reconsolidation of fear.

### The reduction in freezing observed is not due to increases in locomotion

In a separate cohort of mice, dDG cells involved in encoding an acute cocaine exposure were tagged off DOX (Fig. 2i). The next day, to assess whether activation of a cocaine engram would induce hyper-locomotor activity, we tested mice in the open field where we reactivated the cocaine engram in the last 5 min of the 10 min test. We found no group differences in total number of line crossings (Fig. 2j), distance

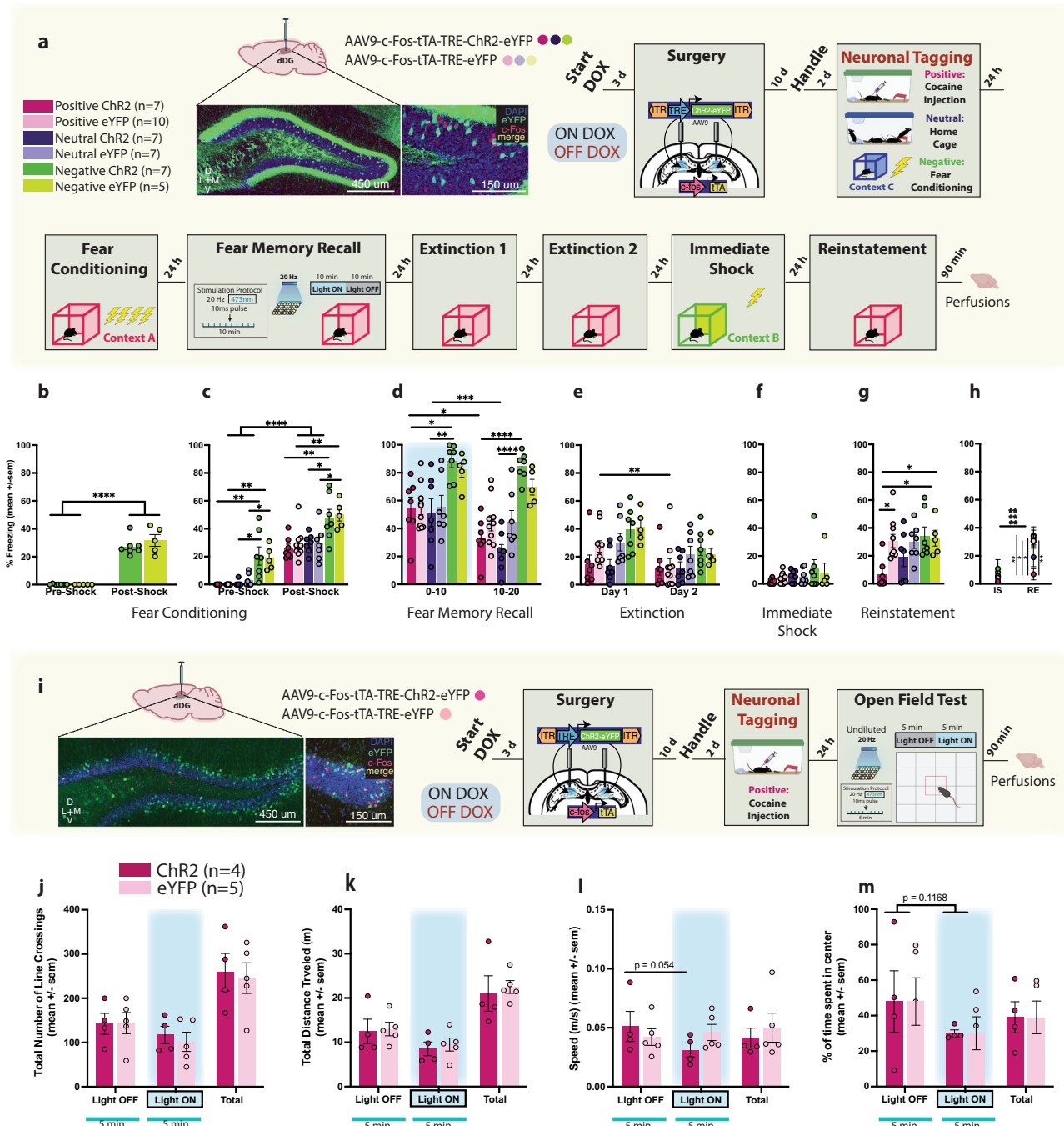

**Fig. 2 | Valence matters: Artificial reactivation of a neutral home cage experience during reconsolidation is not sufficient to disrupt a fear memory. a** Viral strategy and experimental design. dDG cells encoding positive, neutral, and negatively-valenced behavioral epochs were tagged off-DOX (orange).
**b** Negative groups were initially FC in an alternate context (Context C). These mice showed greater freezing post-shock (two-way RM ANOVA: F(1,10) = 168.4, P < 0.0001). **c** The next day all mice were FC. While all mice showed increased freezing post-shock (three-way RM ANOVA: F(1,37) = 295, P < 0.0001), mice in the negative groups demonstrated higher freezing pre-shock (Negative: vs. Positive P < 0.0001, vs. Neutral P < 0.0001) (two way RM ANOVA: F(2,40) = 27.55, P < 0.0001), and post-shock (three-way RM ANOVA: F(2,37) = 25.25, P < 0.0001). **d** During recall, negative groups continued to exhibit increased freezing (three-way RM ANOVA: F(2,37) = 3.286, P < 0.0486, Valence × Virus × Time). Negative-ChR2 mice froze more than neutral-ChR2 mice in the first (P = 0.0086) and last 10 min (P < 0.0001) and more than positive-ChR2 mice at both time points (P = 0.255, P < 0.0001). Positive (P = 0.0072) and neutral (P = 0.0002) ChR2 mice showed faster decline in

freezing compared to eYFP controls. **e** During EXT1, positive and neutral-ChR2 mice froze less than other groups (three-way ANOVA: F(2,37) = 4.107, P = 0.0245, Valence × Day; F(2,37) = 6.841, P = 0.0128, Virus × Day). **f** No group differences observed during IS. **g** During RE, positive ChR2-mice showed reduced fear compared to eYFP controls (P = 0.0249) and both negative-ChR2 (P = 0.0147) and eYFP (P = 0.0498) groups (two-way ANOVA: F(1,37) = 5.923, P = 0.0199, Virus; F(2,37) = 3.440, P = 0.0426). **h** Positive-ChR2 mice froze less than other groups except neutral-ChR2 mice at RE compared to IS (three-way RM ANOVA: F(1,37) = 5.912, P = 0.0200, Day × Virus). **i** Viral strategy and experimental design. dDG cells encoding acute cocaine-exposure (positive) were tagged off-DOX (orange) and 24 h later mice were placed in the OF (10 min) where tagged dDG cells were reactivated in the last 5 min. No group differences in locomotor **j** line crossings **k** distance traveled and **l** speed **m** or anxiety measures (percentage time spent in center). Data represented as means ± s.e.m. *P < 0.05, **P < 0.01, ***P < 0.005, ****P < 0.00. dDG dorsal dentate gyrus, DOX doxycycline, EXT extinction, OF open field, IS immediate shock, RE reinstatement. Source data provided as a Source Data file.

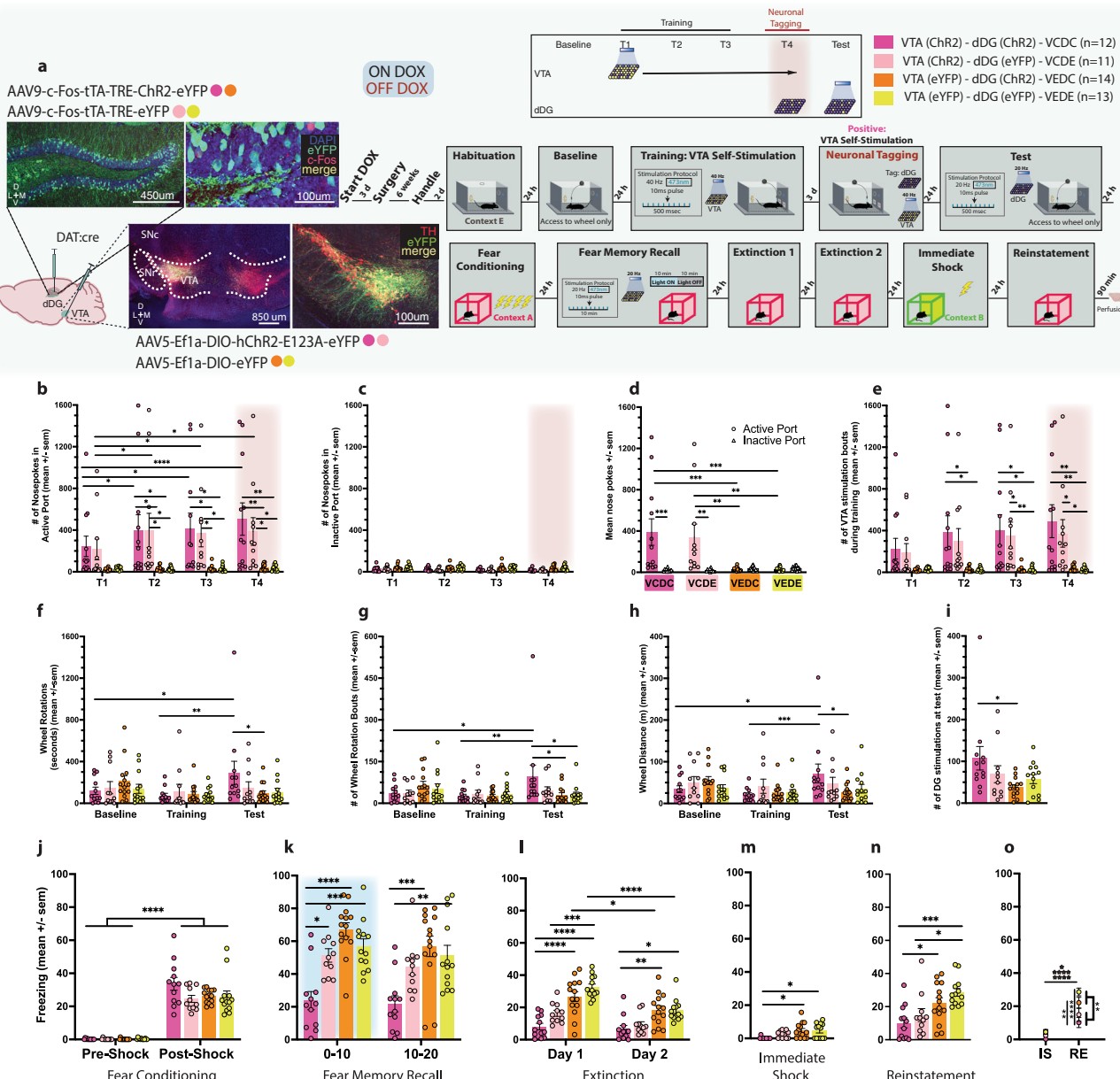

traveled (Fig. 2k), or speed (Fig. 2l) suggesting decreases in freezing observed in the previous experiment were not due to increased loco-motion. Additionally, time spent in the center region revealed no group differences (Fig. 2m) suggesting that artificial activation of a cocaine-related memory is neither anxiogenic, nor anxiolytic.

## Mice will perform an operant response for artificial reactivation of a positive memory

While the hippocampus is implicated in processing positive experiences, it is thought to do so in concert with several regions involved in neuromodulation, including the ventral tegmental area (VTA). The VTA is a critical component of the brain's reward system, and negative affective states (e.g., anxiety) are mediated by VTA dysregulation. It is well established that intracranial self-stimulation (ICSS) of the VTA is a powerfully rewarding operant behavior, where rodents maintain delivery of electrical impulses resulting in dopamine release[54,55]. This procedure has been previously adapted[56–61] to incorporate in vivo optogenetic stimulation of dopaminergic neurons in the VTA. To tag and reactivate dDG cells active during this positive experience, we selectively expressed ChR2 in dopaminergic VTA cells using transgenic

mice which express Cre under control of the dopamine transporter (DAT). We injected our viral vectors AAV5-Ef1a-DIO-(hChR2-E123A)-eYFP and implanted an optic fiber unilaterally into the VTA. We also injected c-Fos-tTA-TRE-(ChR2)-eYFP and implanted optic fibers bilaterally aimed at the dDG (Fig. 3a). Mice were initially habituated to the operant chamber and given access to a wheel with no consequences, which served as a baseline measure. The following day, mice were placed back into the operant box, and two nose ports were introduced, one active and one inactive. Nose pokes into the active port produced optogenetic VTA stimulation, while nose pokes into the inactive port produced no stimulation and served as a discriminative control. Mice were given three ICSS training sessions and then taken off DOX. They were brought back for a fourth training session, in which dDG cells responsive to VTA self-stimulation were tagged. The following day, access to the nose ports was restricted, and wheel spins produced optical dDG stimulation to reactivate the VTA self-stimulation engram. This was done to assess whether mice would perform an operant response for a positive (VTA-ChR2, dDG-ChR2) or neutral (VTA-eYFP, dDG-ChR2) experience compared to dDG-eYFP controls (Fig. 3a). In mice injected with ChR2 in the VTA, nose pokes into the active port

**Fig. 3 | Mice will perform an operant response for artificial reactivation of a positive memory. a** Viral strategy and experimental design. DAT-Cre mice were given access to a wheel that produced no consequences (baseline and training). They were then trained to nose poke for 500 ms bursts of optogenetic VTA stimulation (T1–T3). dDG cells encoding this positive experience were tagged off-DOX (orange, T4). The next day, mice were again given access to the wheel, where spinning it now produced artificial reactivation of the tagged dDG cells responsive to VTA self-stimulation. 24 h later, mice underwent the same experimental procedure as previous experiments. Mice were injected with either ChR2 (VC) or eYFP (VE) in the VTA, and also injected with either ChR2 (DC) or eYFP (DE) in the dDG. **b** In the active port, VCDC & VCDE mice nose-poked more than VEDC & VEDE mice (two-way RM ANOVA: $F_{(9,138)} = 2.179$, $P = 0.0270$, Time × Group) and increased responding across days whereas **c**, no effects were seen for the inactive port. **d** Summary of nose poke behavior across days (two-way RM ANOVA: $F_{(3,46)} = 6.102$, $P = 0.0014$, Noseport × Group). **e** Active port nose pokes resulted in VTA stimulation bouts, which were observed significantly more in VCDC and VCDE groups (two-way RM ANOVA: $F_{(9,138)} = 2.472$, $P = 0.0120$, Time × Group). **f** We found VCDC mice spun the wheel longer during the test compared to baseline ($P = 0.0230$) and training ($P = 0.0029$) and also longer than VEDC mice during the test ($P = 0.0264$), demonstrating they will perform an operant response for access to a positive memory (two-way RM ANOVA: $F_{(6,92)} = 2.233$, $P = 0.0467$, Time × Group). **g** We found a similar result for number of wheel spins (At test - VCDC vs. VEDC: $P = 0.0108$, VCDC vs. VEDE: $P = 0.0133$; VCDC - from baseline to test: $P = 0.0148$, from training to test: $P = 0.0032$) (two-way RM ANOVA: $F_{(6,92)} = 2.513$, $P = 0.0269$, Time × Group). **h** and wheel distance (At test - VCDC vs. VEDC:

$P = 0.0246$; VCDC - from baseline to test: $P = 0.0177$, from training to test: $P = 0.0009$) (two-way RM ANOVA: $F_{(6,92)} = 2.575$, $P = 0.0237$, Time × Group). **i** VCDC mice produced more dDG stimulations at test than VEDC mice ($P = 0.0318$) (one-way ANOVA: $F_{(3,46)} = 2.836$, $P = 0.0484$). **j** Mice showed post-shock freezing following FC (three way RM ANOVA: $F_{(1,92)} = 386.4$, $P < 0.0001$, Time). **k** During recall, stimulation produced decreases in freezing in the VCDC group compared to other groups (vs. VCDE: $P = 0.0112$, VEDC: $P < 0.0001$, VEDE: $P = 0.0004$) persisting into the last half of the session (vs. VEDC: $P = 0.0001$, VEDE: $P = 0.0027$) (three-way RM ANOVA: $F_{(1,46)} = 11.65$, $P = 0.0013$, dDG Virus × VTA Virus; $F_{(1,46)} = 9.841$, $P = 0.003$, Time). **l** During EXT, VCDC mice showed less freezing compared to VEDC (Day 1: $P < 0.0001$, Day 2: $P = 0.0090$) and VEDE (Day 1: $P < 0.0001$, Day 2: $P = 0.0144$) controls (three way RM ANOVA: $F_{(1,46)} = 8.526$, $P = 0.0054$, VTA Virus × Day; $F_{(1,46)} = 5.896$, $P = 0.0191$, dDG Virus × Day). **m** These same differences were seen during IS (VCDC vs. VEDC: $P = 0.0107$; VCDC vs. VEDE: $P = 0.0201$) (two-way ANOVA: $F_{(1,46)} = 10.70$, $P = 0.002$, VTA Virus). **n** During RE, VCDC mice showed reduced fear compared to VEDC ($P = 0.0405$) and VEDE controls ($P = 0.0007$). VCDE mice, which differed from VEDE mice by their VTA self-stimulation experience alone, also showed less freezing ($P = 0.0199$) (two-way ANOVA: $F_{(1,46)} = 16.78$, $P = 0.0002$, VTA Virus). **o** VCDC and VCDE mice froze less than VTA-eYFP controls (VEDC & VEDE mice) at RE (three-way RM ANOVA: $F_{(1,46)} = 7.879$, $P = 0.0073$, VTA Virus × Day). Data represented as means ± s.e.m. $*P < 0.05$, $**P < 0.01$, $***P < 0.005$, $****P < 0.00$. dDG dorsal dentate gyrus, DOX doxycycline, EXT extinction, IS immediate shock, RE reinstatement, VTA ventral tegmental area. Source data provided as a Source Data file.

were significantly higher than the inactive port, and they increased across sessions demonstrating the mice's ability to discriminate between ports and self-deliver optical stimulation for reward (Fig. 3b–e). Comparing wheel baseline measures to training and test, mice injected with ChR2 in the VTA and dDG, completed more wheel rotations, which were kept in motion for longer durations and distances (Fig. 3f–h) and produced more stimulations (Fig. 3i) compared to all other groups. This finding demonstrates that mice will perform an operant response to maintain artificial reactivation of a positive memory, specifically the memory of VTA self-stimulation. Mice did not exhibit this behavior for a memory of operant box exposure in the absence of VTA stimulation. Following this test, mice underwent the same experimental protocol as before where they were FC in context A where they demonstrated post-shock freezing (Fig. 3j and Supplementary Fig. 1e) and then given a fear memory recall test. Reactivating the VTA self-stimulation engram (VCDC) during recall reduced freezing throughout the session (Fig. 3k). Levels remained low throughout extinction (Fig. 3l and Supplementary Fig. 1e), immediate shock (Fig. 3m), and reinstatement (Fig. 3n). Interestingly, between the two dDG-eYFP groups, the group that had received VTA stimulation earlier (VCDE) demonstrated a beneficial effect of this experience exhibiting intermediate levels of freezing compared to the VCDC group and the other control groups on EXT1 (Fig. 3l), and from immediate shock to reinstatement (Fig. 3o). Together, these results show that interference from a rewarding experience can counteract negative affective states.

### Activation of randomly labeled dDG neurons is also sufficient to promote the reconsolidation of fear

Next we asked, if artificial reactivation of a positive memory, which involves stimulation of a small set of neurons (<10%)[38], can update a fear memory during reconsolidation, could we circumvent the positive-valence prerequisite to achieve a similar effect if we activate a larger population of neurons not necessarily tied to a memory? Unlike previous experiments, where we used a cFos-inducible tagging strategy to label cells involved in different experiences, here, we used a virus with a constitutive promoter (CaMKIIa) to randomly tag dDG neurons with ChR2 (Fig. 4a). Mice were injected with either undiluted or diluted virus to label a large percentage or fraction of dDG cells, respectively. Mice demonstrated post-shock freezing after being FC (Fig. 4b and Supplementary Fig. 1f). During recall the next day, the

labeled neurons were optically stimulated. During the first half of the session, we saw real-time decreases in freezing in both undiluted and diluted-ChR2 groups but by the second half, only the undiluted group showed less freezing (Fig. 4c). The undiluted-ChR2 group continued to exhibit less freezing throughout extinction (EXT1 & EXT2) (Fig. 4d) and both undiluted and diluted ChR2 groups froze less than the eYFP groups during the first 3 min of EXT1 (Supplementary Fig. 1f). As expected, there were no group differences during immediate shock (Fig. 4e). At reinstatement, we observed reduced freezing in both undiluted and diluted-ChR2 groups compared to eYFP controls (Fig. 4f) and compared to immediate shock (Fig. 4g). Our effects were greater in the undiluted group, suggesting reconsolidation-based processes can be potentially engaged by activating ensembles that are not connected to an engram of a particular valence if enough cells are activated. This memory modulation strategy may be akin to stimulation protocols currently approved for use in humans[62,63].

### dDG interference is specific to the fear memory and does not affect other hippocampal-mediated memories

To determine whether our manipulation, which disrupts the fear memory, would also indiscriminately affect other types of hippocampal-mediated memory, we trained mice on a spatial reference memory task. Like the previous experiment, we injected mice with undiluted AAV5-CaMKIIa-(hChR2-H134R)-eYFP to randomly label dDG neurons (Fig. 5a). Water-restricted mice were then trained to obtain a water reward from one of the arms in an 8-arm radial maze. They received 4 trials/d until a set of performance criteria were met. Afterwards, to assess the effect of dDG stimulation on the fear memory, they underwent the same behavioral protocol as previous experiments. To assess effects of dDG stimulation on the spatial reference memory, we retested mice on spatial memory performance after fear-conditioning and after recall and ran a curtain probe trial after EXT2 to ensure mice were using extra-maze cues to perform the task. Interestingly, mice in this experiment demonstrated lower post-shock freezing during fear-conditioning compared to other mice in the study, which we believe is a direct result of being handled more often due to training on the spatial reference task (Fig. 5b and Supplemental Fig. 1g). Nonetheless, during recall, eYFP mice showed normal freezing whereas as expected, ChR2-mice showed real-time decreases in freezing (Fig. 5c). This diminished fear responding persisted during

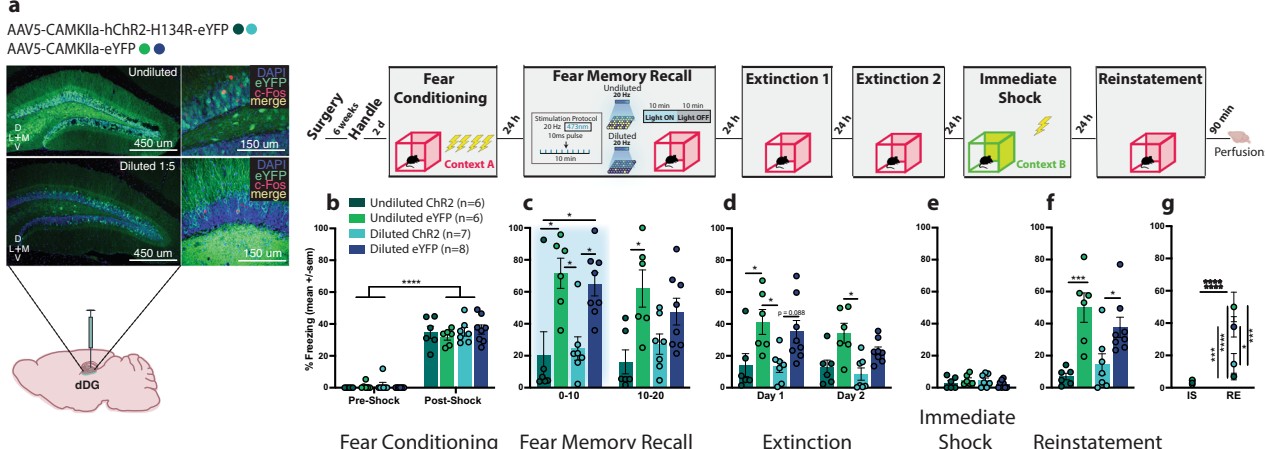

**Fig. 4 | Activation of randomly labeled dDG is also sufficient to disrupt fear reconsolidation. a** Viral strategy and experimental design. We used a virus with a constitutive promoter to randomly label dDG neurons not tied to the encoding of a behavioral epoch. Mice were injected with either undiluted or diluted virus. The experimental procedure was similar to the previous experiments with the exception of the neuronal tagging component. **b** During FC, all mice froze post-shock (three-way RM ANOVA: $F_{(1,23)} = 447.9$, $P < 0.0001$). **c** During recall, stimulation produced decreases in freezing in both the undiluted ($P = 0.0106$) and diluted ($P = 0.0369$) ChR2-groups compared to eYFP-controls (three-way RM ANOVA: $F_{(1,23)} = 25.67$, $P < 0.0001$, Virus). In the latter half of the session, mice in the undiluted ChR2 continued to show decreased freezing compared to eYFP-controls

($P = 0.0300$). **d** During EXT, there was an overall decrease in freezing across days (three-way RM ANOVA: $F_{(1,23)} = 4.312$, $P = 0.0492$) and a reduction in freezing in ChR2-groups compared to eYFP-controls (three-way ANOVA: $F_{(1,23)} = 20.88$, $P = 0.0001$). More specifically, in the undiluted groups on day 1 ($P = 0.0394$). **e** No group differences were seen during IS. **f** During RE, mice in both the undiluted ($P = 0.0004$) and diluted ($P = 0.0304$) ChR2-groups froze less compared to eYFP-controls (two-way ANOVA: $F_{(1,23)} = 25.15$, $P < 0.0001$). **g** They also froze less at RE compared to IS (three-way RM ANOVA: $F_{(1,23)} = 27.08$, $P < 0.0001$, Virus × Day). Data represented as means ± s.e.m. *$P < 0.05$, **$P < 0.01$, ***$P < 0.005$, ****$P < 0.00$. dDG dorsal dentate gyrus, EXT extinction, IS immediate shock, RE reinstatement. Source data are provided as a Source Data file.

extinction (Fig. 5d) with ChR2 mice showing less freezing in the first 3 min of EXT1 (Supplemental Fig. 1g) and throughout the entire EXT2 session. No differences were seen during immediate shock (Fig. 5e), and as expected, ChR2-groups demonstrated less freezing during reinstatement (Fig. 5f, g). For maze performance, all mice met criteria between 5 and 14 days (Fig. 5h) and improved across time during acquisition, in terms of latency to find the reward, number of arm-deviations (upon a mouse's first visit to an arm), number of reference errors (i.e., entering wrong arm) and repeated reference errors (working memory errors) made (Fig. 5i–p, left-most panels). Performance was divided into two categories: trial 1, which was interpreted as an assessment of long-term memory from the day before, and trials 2–4, which were interpreted as an assessment of short-term memory from the previous trial on the same day. Mice were first retested in the maze 3 h after fear-conditioning to confirm that fear-conditioning itself did not affect spatial memory. We saw no effects (Fig. 5i–p, middle panels).

Working under the assumption that disruption of the fear memory by our optical intervention occurs because conditions are present for memory to undergo reconsolidation, observing an impairment of spatial memory could be obscured if this memory does not also undergo reconsolidation. To accurately test whether our intervention was specific to the fear memory, we sought to increase the likelihood that boundary conditions permitting memory reconsolidation for the spatial memory were met[64]. To that end, we gave half the mice (Reminder groups) one reminder session (trial 1) 5 min prior to the fear memory recall test. During this reminder session, we replaced the flooring in the maze center with sandpaper to create a prediction error[65]. This did not affect reward location, but it did present mice with an unexpected cue during the trial. Reminder-mice received trials 2–4 3 h after the recall session, while the other half of mice (No Reminder groups) did not receive a reminder session, and instead received all 4 regular trials 3 h after the recall session. Again, we saw no differences in performance and all mice performed well within the range of criterion (Fig. 5i–p, middle panels). All mice were tested on the maze the following day and 3 h later were given the first extinction session. No

measures were affected during this test except arm-deviations (Fig. 5k, l). On trial 1, both ChR2 and eYFP mice in the Reminder groups had higher arm-deviations suggesting the reminder session, which potentially led to reconsolidation of the spatial memory, briefly affected this measure of performance. However, this effect was behavioral and not a result of optical stimulation of random dDG cells as there were no differences between ChR2 and eYFP mice. On trials 2–4, mice in ChR2 groups (both Reminder and No Reminder) demonstrated better performance (fewer arm-deviations) compared to eYFP groups suggesting maze performance improved as a result of reconsolidation regardless of whether a reminder session was given. Importantly, this also demonstrated that the disruptive effect of our manipulation was specific to the fear memory (Fig. 5i–p, middle panels).

To ensure we were testing hippocampal-mediated memory where mice were using extra-maze cues to find the reward, the following day mice were given a curtain probe test. As expected, all mice showed decreased performance on the probe compared to the last 3 d of training with performance reaching the level it was at when they first started the task (Fig. 5i–p, right-most panels).

**Rewriting the original fear memory**
Next, we sought to determine whether the reduction in conditioned fear was accompanied by a change in ensemble dynamics of the original fear engram. To assess whether our manipulation had altered the original fear memory, we combined two virus-based systems (Fig. 6a). All mice were injected with c-Fos-tTA-TRE-mCherry to tag dDG cells involved in encoding the fear-conditioning epoch. Mice were also injected with either undiluted AAV5-CaMKIIa-hChR2-H134R-eYFP or undiluted AAV5-CaMKIIa-eYFP to label a competing set of dDG neurons. Mice then underwent the same experimental protocol as before where they were FC demonstrating post-shock freezing (Fig. 6b and Supplemental Fig. 1h). During recall, optical stimulation produced real time decreases in freezing in ChR2 mice compared to eYFP controls (Fig. 6c), and these differences persisted in extinction (Fig. 6d) including the first 3 min of EXT1 and the last 3 min of EXT2 (Supplemental Fig. 1h). No group differences were observed during immediate

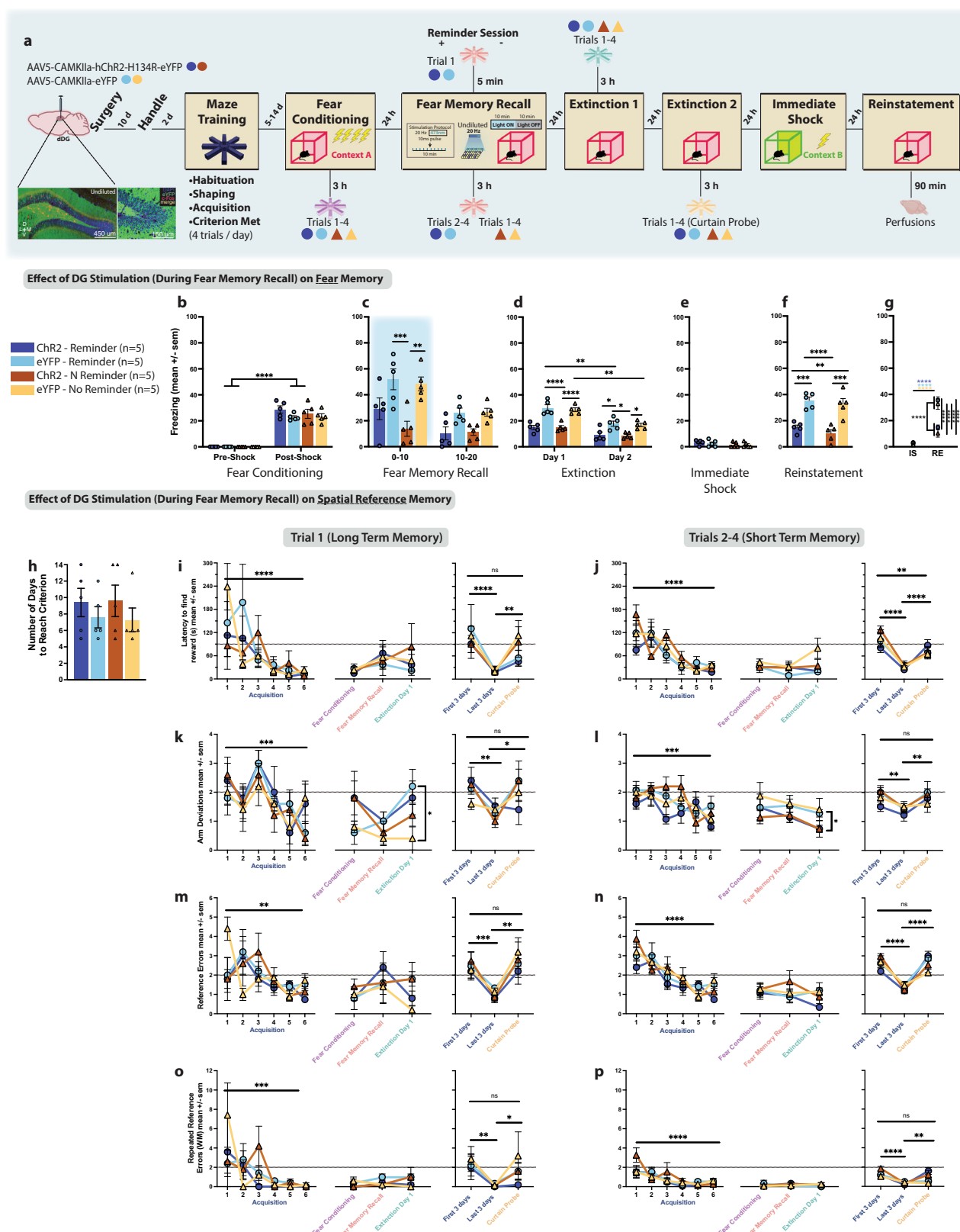

shock (Fig. 6e). During reinstatement, ChR2 mice continued to show decreased freezing (Fig. 6f, g). To determine whether these mice had fewer overlapping neurons from the original fear-conditioning epoch to the reinstatement test compared to eYFP controls, mice were perfused 90 min after reinstatement, and c-Fos levels were quantified.

For each experiment, representative images from each neuronal tagging strategy were taken (Fig. 7a–d and Supplemental Fig. 5a–e) and used to obtain cell counts. We first calculated the total number of DAPI labeled cells for each group (Fig. 7e, f). For experiments where we tagged a behavioral epoch using a c-Fos promoter (Figs. 1–3, 6), the size of the engram was determined as a percentage of DAPI-labeled

**Fig. 5 | dDG interference is specific to the fear memory and does not affect other hippocampal-mediated memories. a** Viral strategy and experimental design. We randomly labeled a dDG ensemble and then trained water-restricted mice to obtain water from one of the arms in an 8-arm radial maze (4 trials/day). Following criterion, they were FC and 3 h later tested on spatial memory. The next day they received a recall test during which labeled cells were stimulated. Half the mice received one reminder session (trial 1, T1) 5 min prior to recall, and trials 2–4 (T2–4) 3 h after recall. The other half did not receive a reminder session, and instead all 4 regular trials 3 h after recall. Spatial performance was tested the next day, 3 h before the first EXT session. The next day mice got a second EXT session and 3 h later a curtain probe. The next day, mice underwent IS and 24 h later a RE test. Effect of dDG Stimulation on Fear Memory: **b** During FC, all mice froze post-shock (three-way RM ANOVA: F(1,16) = 377.1, P < 0.0001). **c** During recall, stimulation decreased freezing in the No Reminder-ChR2 group (P = 0.0027) compared to eYFP controls (three-way RM ANOVA: F(1,16) = 26.11, P = 0.0001, Time; F(1,16) = 24.75, P = 0.0001, Virus). **d** Freezing decreased across EXT days and in ChR2 groups compared to eYFP controls (three-way RM ANOVA: F(1,16) = 6.008, P = 0.0261, Time × Virus). More specifically, in No Reminder groups on day 1 (P < 0.0001) and in both Reminder (P = 0.0279) and No Reminder (P = 0.0395) groups on day 2. **e** No group differences seen during IS. **f** During RE, mice in both Reminder (P = 0.0008) and No Reminder (P = 0.0003) ChR2 groups froze less compared to eYFP controls (two-way ANOVA: F(1,16) = 53.27, P < 0.0001). **g** and at RE compared to IS (three-way RM ANOVA: F(1,16) = 66.81, P < 0.0001, Virus × Day). Effect of dDG Stimulation on the Spatial Memory: **h** Mice reached criterion in 5–14 days. Dependent measures: **i, j** latency to find reward, **k, l** arm-deviations, **m, n** reference errors, and **o, p** repeated reference errors. During acquisition (left panels, first 3 and last 3 days of training), for T1 (long-term memory) and T2–4 (short-term memory) all mice

improved across training (three-way RM ANOVAs. Latency - T1: F(5,48) = 8.053, P < 0.0001, T2–4: F(5,48) = 19.66, P < 0.0001; Arm-Deviations - T1: F(5,48) = 6.817, P < 0.0001, T2–4: F(5,48) = 5.186, P = 0.0007; Reference Errors - T1: F(5,48) = 3.921, P = 0.0046, T2–4: F(5,48) = 14.94, P < 0.0001; Repeated Reference Errors - T1: F(5,48) = 6.392, P = 0.0001, T2–4: F(5,48) = 9.848, P < 0.0001). We compared spatial performance after FC (purple), before (Reminder groups only) and after recall (pink), and before EXT1 (mint) (middle panels). We only saw differences for arm-deviations. For T1, both ChR2 and eYFP-mice in the Reminder groups had higher arm-deviations during the first trial prior to EXT1 (three-way RM ANOVA: F(1,24) = 4.741, P = 0.0395). As differences were not between ChR2 and eYFP mice, this suggests the Reminder session, and not the reconsolidation-based manipulation, briefly affected this measure. Briefly since, on T2–4: mice in the ChR2 groups (Reminder and No Reminder) demonstrated better performance (fewer arm-deviations) compared to eYFP groups on EXT 1, suggesting the manipulation improved performance on the maze despite whether a reminder session was given (three-way RM ANOVA: F(1,24) = 6.819, P = 0.0153). For the curtain probe, extra-maze cues were not visible. We compared performance on this test to the first and last 3 days of acquisition (peach) (right panels). All mice performed as poorly as the start of training (three-way RM ANOVAs. Latency - T1: F(2,24) = 15.16, P < 0.0001, T2–4: F(2,24) = 43.93, P < 0.0001; Arm-Deviations - T1: F(2,24) = 5.671, P = 0.0096, T2–4: F(2,24) = 8.610, P = 0.0015; Reference Errors - T1: F(2,24) = 13.80, P = 0.0001, T2–4: F(2,24) = 6.009, P < 0.0077, Time × Reminder; Repeated Reference Errors - T1: F(2,24) = 7.927, P = 0.0023, T2–4: F(2,24) = 5.862, P = 0.0085, Time × Virus). Dotted lines: criterion required for 2 consecutive days. Data represented as means ± s.e.m. *P < 0.05, **P < 0.01, ***P < 0.005, ****P < 0.00. dDG dorsal dentate gyrus, CP curtain probe, EXT extinction, F3 first 3 days, IS immediate shock, L3 last 3 days, RE reinstatement. Source data provided as a Source Data file.

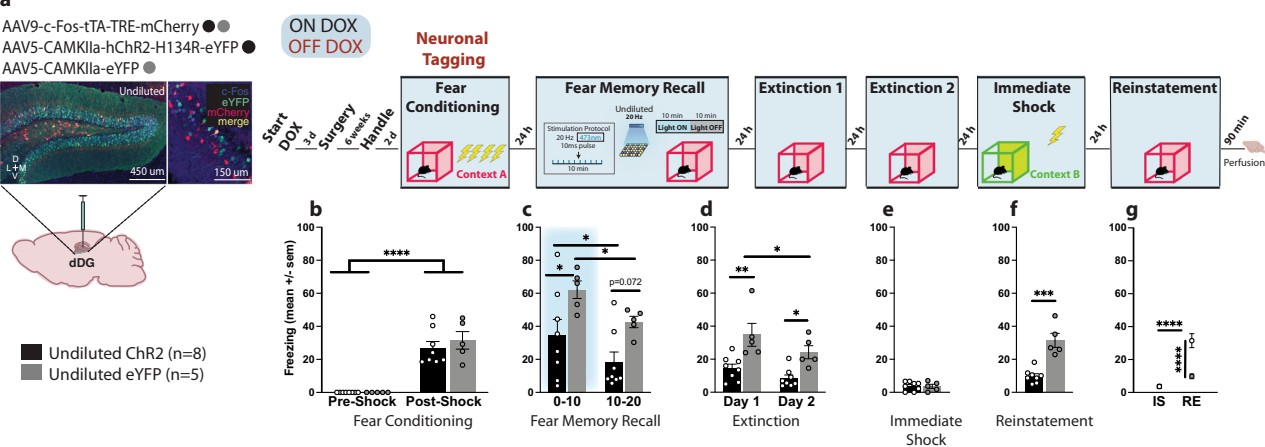

**Fig. 6 | A tale of two memories. a** Viral strategy and experimental design. To assess whether our reconsolidation-based manipulation was able to alter the original fear engram, we combined two viral strategies. All mice were injected with c-Fos-tTA-TRE-mCherry to tag dDG cells encoding the FC epoch. Mice were also injected with either undiluted AAV5-CaMKIIa-hChR2-H134R-eYFP or undiluted AAV5-CaMKIIa-eYFP to randomly label dDG neurons. Mice then underwent the same experimental protocol as before. **b** All mice froze post-shock (two-way RM ANOVA: F(1,11) = 85.26, P < 0.0001). **c** During recall, stimulation produced decreases in freezing (0–10) (P = 0.0392) in ChR2-mice compared to eYFP controls (two-way RM ANOVA: F(1,11) = 20.32, P = 0.0009, Time; F(1,11) = 6.468, P = 0.0273, Virus).

**d** These differences persisted in EXT (Day 1: P = 0.0016; Day 2: P = 0.0127) (two-way RM ANOVA: F(1,11) = 15.05, P = 0.0026, Time; F(1,11) = 14.26, P = 0.0031, Virus). **e** No group differences were observed during IS. **f** During reinstatement, mice in the ChR2 group continued to show decreased of freezing (Unpaired t-test: t(11) = 5.768, P = 0.0001, two-tailed). **g** They also froze less at RE compared to IS (two-way RM ANOVA: F(1,11) = 22.40, P = 0.0006, Virus × Day). Data represented as means ± s.e.m. *P < 0.05, **P < 0.01, ***P < 0.005, ****P < 0.00. dDG: dorsal dentate gyrus, DOX doxycycline, EXT extinction, IS immediate shock, RE reinstatement. Source data are provided as a Source Data file.

neurons (Fig. 7g, eYFP; Fig. 7h, mCherry). Regardless of valence, tagged cells associated with a particular behavioral experience, including a home cage experience[38], consisted of approximately 8% of DAPI-labeled neurons in dDG (M = 8.015, SD = 0.572). The percentage of cells labeled as c-Fos[+], representing the set of cells active during the reinstatement test (Fig. 7g, i, RFP; Fig. 7h, j, RFP & BFP) was approximately 1.5% (M = 1.438, SD = 0.21). In the first set of experiments, the percentage of overlap (yellow) (Fig. 7g, k) between the tagged engram used to interfere with the fear memory during recall (eYFP) (Fig. 7g) and the engram at reinstatement (RFP) (Fig. 7g, i) can be interpreted as a

measure of whether our reconsolidation-based manipulation resulted in merging the two memories together. We observed a greater percentage of overlap for negative experiences compared to neutral experiences suggesting the negative experiences shared higher similarity with the fear memory compared to the neutral experiences. However, no differences were observed between the percentage of overlap for negative experiences compared to positive experiences, or positive experiences compared to neutral experiences. While we speculate that the degree of overlap between negative experiences and the fear memory may be preventing reactivation of those engrams

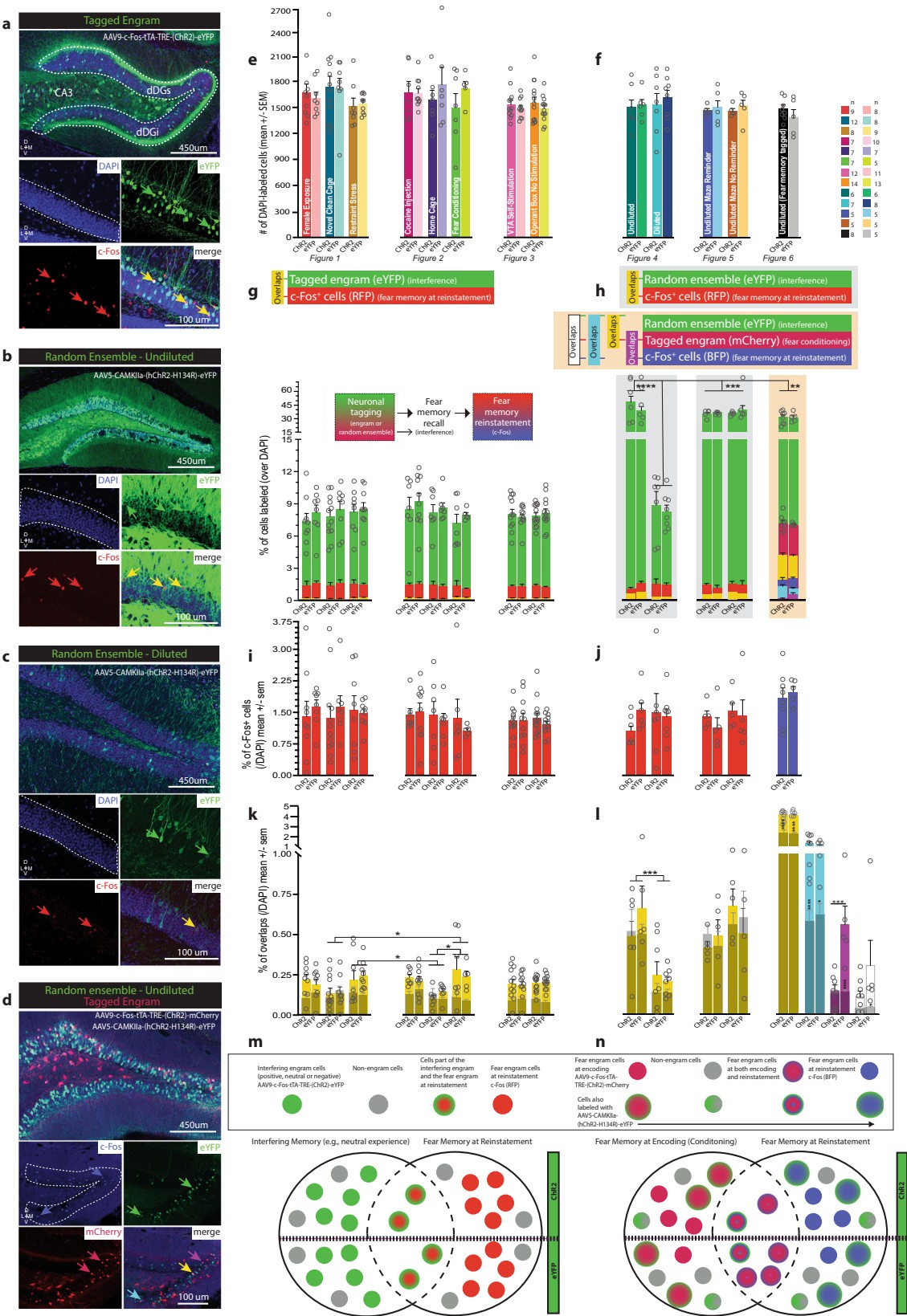

from interfering with the fear memory, we cannot assume that our ability to successfully weaken a fear memory by reactivating a positive memory derives solely from a disparity in these populations.

There were no differences between ChR2 and eYFP mice, leading us to conclude that stimulation of an interfering engram during recall did not increase the similarity between the engram and the fear

memory at reinstatement (Fig. 7k). Similarly, when we tagged neurons not involved in the encoding of a behavioral epoch, we saw a similar trend where overlaps did not differ between ChR2 and eYFP groups suggesting that activation of the randomly labeled neurons did not then become recruited into the fear memory engram at reinstatement (Fig. 7h, l). However, there was a higher degree of overlap in the

**Fig. 7 | Perturbing dDG neurons during fear memory reconsolidation rewrites original fear memory but does not bias it in the direction of the interfering cellular ensemble.** Representative dDG images (20×) of **a** a tagged engram (dDGs: suprapyramidal layer; dDGi: infrapyramidal layer). **b** Randomly-labeled ensemble-undiluted. **c** -diluted. DAPI (blue, enclosed within white dotted lines), eYFP (green), c-Fos (red), overlaps (yellow). **d** Both, a tagged engram (mCherry, red) and randomly-labeled ensemble (undiluted) (eYFP, green). c-Fos (blue), mCherry-eYFP overlaps (yellow), mCherry-c-Fos overlaps (magenta), eYFP-c-Fos overlaps (cyan). **e** DAPI-labeled cells across experiments—where engrams were tagged. **f** —where ensembles were labeled randomly. **g** Percentage of cells (/DAPI) labeled with eYFP (green, tagged engram), RFP (c-Fos$^+$, red, fear memory at RE), or both (overlaps, yellow) corresponding to graph above. **h** Left & Middle: Percentage of cells (/DAPI) labeled with eYFP (green, random ensemble), RFP (c-Fos$^+$, red, fear memory at RE), or both (overlaps, yellow) corresponding to graph above. Percentage of dDG cells labeled with diluted virus similar to size of tagged engram and significantly lower than dDG cells labeled with undiluted virus (two-way ANOVA: F(4,54) = 27.55, $P < 0.0001$). Right: Percentage of cells (/DAPI) labeled with eYFP (green, random ensemble), mCherry (tagged FC engram), or both (overlaps, yellow), BFP (c-Fos$^+$, blue, fear memory at RE), eYFP and BFP (overlaps, cyan), and mCherry and BFP (overlaps, magenta) corresponding to graphs above. **i, j** Percentage of c-Fos$^+$ cells (/DAPI) labeled with RFP or BFP enlarged. **k** Percentage of overlaps (/DAPI) corresponding to graphs above and set against chance (grey). Higher overlap between fear memory at RE and engrams associated with negative experiences compared to engrams associated with neutral experiences despite viral condition (restraint stress vs. home cage: $P = 0.0461$, FC vs. home cage: $P = 0.0121$, FC vs. novel clean cage: $P = 0.0018$) (two-way ANOVA: F(7,131) = 3.427, $P = 0.0021$). Overlaps not significantly higher than chance suggesting manipulation did not bias fear engram toward interfering engram even when fear was significantly reduced. **l** Percentage of overlaps (/DAPI) corresponding to graphs above, enlarged and set against chance (grey). Left & Middle: Overlaps higher when mice injected with undiluted compared to diluted virus (three-way RM ANOVA: F(1,26) = 7.356, $P = 0.0117$, Dilution × Chance), a direct result of more cells labeled. Not significantly greater than chance suggesting randomly labeled cells did not disproportionately include fear engram RE cells involved (those cells did not become ensuing fear engram). Right: Percentage of cells part of original fear memory and part of fear memory at RE significantly lower in ChR2-mice (magenta). Reconsolidation-based manipulation caused orthogonal disengagement of these ensembles, separating memories to degree of overlap expected in differentially-valenced memories ($P = 0.0005$) or simply chance levels ($P < 0.0001$) (two-way RM ANOVA: F(1,11) = 15.70, $P = 0.0022$, Virus × Chance). Proportion of randomly labeled cells/activated (green) also part of original fear engram (cherry) was above chance (yellow, ChR2: $P < 0.0001$, eYFP: $P < 0.0001$) (two-way RM ANOVA: F(1,11) = 155.3, $P < 0.0001$). However, proportion of cells randomly labeled/activated (green) also part of the fear engram at RE (blue) was also above chance (cyan, ChR2: $P < 0.0001$, eYFP: $P = 0.0103$) (two-way RM ANOVA: F(1,11) = 48.46, $P < 0.0001$. **m, n** Schematic depicting ensemble dynamics described in **k, l**. *N* values listed in panel (**f**). Data represented as means ± s.e.m. *$P < 0.05$, **$P < 0.01$, ***$P < 0.005$, ****$P < 0.00$. dDG: dorsal dentate gyrus, FC fear conditioning, RE reinstatement. Source data provided as a Source Data file.

---

undiluted groups compared to the diluted groups based simply on the number cells tagged. In the undiluted groups, we tagged approximately 40% ($M = 37.16$, SD = 5.5) of DAPI-labeled neurons and in the diluted groups we tagged a similar 8% of cells ($M = 8.57$, SD = 0.4) as our epoch-associated engrams (Fig. 7h). Finally, we sought to determine whether the original fear memory was changed from conditioning to reinstatement given the reduction in fear, despite not being biased to incorporate the cells artificially activated during recall. The percentage of overlaps (magenta) (Fig. 7h, l) between these sets of neurons, those tagged in the original fear engram (mCherry) (Fig. 7h) and the fear engram at reinstatement (BFP) (Fig. 7h, j) can be interpreted as a measure of whether our reconsolidation-based intervention resulted in alteration of the original fear memory. We found significantly fewer overlaps in the fear memory across conditioning and reinstatement in the ChR2 group, suggesting that while we did not merge the fear memory with the interfering ensembles (Fig. 7m), our manipulations, nonetheless, altered the original fear memory (Fig. 7n), which we believe may help explain corresponding reductions in fear. We speculate that our reconsolidation-based manipulation caused a disengagement of these ensembles in an orthogonal manner, separating these memories into disparate neuronal populations (Fig. 7m, n). Interestingly, the proportion of randomly labeled/activated (eYFP) cells that were also part of the original fear engram (mCherry) was above chance (yellow) and the proportion of randomly labeled/activated (eYFP) cells that were also part of the fear engram at reinstatement (BFP) was also above chance (cyan) as was overlap of all 3 types of cells (white) in the eYFP group (Fig. 7l); however, no group differences were seen. Taken together, these findings provide preliminary evidence for the potential therapeutic efficacy of artificially modulating memories to both acutely and enduringly suppress fear responses by altering the original fear memory during the reconsolidation period.

## Discussion

Here, we combined our viral neuronal tagging strategy with optogenetics to manipulate hippocampal ensembles and disrupt the expression of a fear memory in mice. We showed that hippocampal interference induced by optical reactivation of a competing, positive memory was sufficient to update the fear memory during natural reactivation in a recall session through reconsolidation. These effects were not due to the viral injections themselves nor the light stimulation alone and were only observed when the positive memory was reactivated in the conditioning context.

While it is generally considered evolutionarily advantageous to remember emotionally significant events well, the pursuit of therapeutic forgetting has emerged in cases such as PTSD where these memories become debilitatingly intrusive. Currently, the majority of pharmacological and cognitive behavioral treatments used to treat disorders of emotional memory typically only affect the strength of the affective response while the original fear memory is left intact[66]. As a result, these memories often recover their strength following subsequent aversive events involving stress. Thus, there is an urgent need for a shift toward mechanism-guided psychiatric therapeutics[67]. Studies testing reconsolidation theory have provided experimental evidence that memories are not immutable and can be updated if certain conditions are met[7,68]. One condition is that the memory must be reactivated, and secondly, that the treatment aimed at altering the memory must happen post-reactivation[69]. Our reconsolidation-based manipulation, aimed at updating the original fear engram with attributes of a positively valenced, tagged memory, allowed for both these requirements to be satisfied. However, to limit the optical stimulation period, we designed our first experiment to test whether the stimulation protocol would be more effective when administered in the first or last half of the session. We originally hypothesized that the manipulation would be more effective when administered in the latter half of the session (L10) since the memory would have already been online, satisfying this condition for reconsolidation. We were surprised to find our manipulation worked more effectively when it occurred simultaneously with memory recall (F10) as opposed to ten minutes post-reactivation. We speculate that the efficacy of our stimulation may be disrupted when it occurs later in the session as the mice are taken out of the chamber as soon as stimulation ends. We believe that this type of stimulation may promote new learning[70] (e.g., extinction) and since the mice were left in the chamber for an additional ten minutes in the F10 condition, the context could theoretically serve as a substrate for learning, especially given that conditioned fear decreases across the session. However, it is also possible that since mice were shocked within the interval of 198–440 s during fear-conditioning the prior day, that the memory was more malleable early in the recall session as this is when the highest degree of shock expectation would theoretically occur. This could explain why the fear memory may have become

**Fig. 8 | The significance of valence. a** Reactivation of a competing positive memory via optical stimulation is sufficient to update a fear memory during reconsolidation. Reactivation of an engram associated with a negative experience, or a neutral experience with the exception of exposure to a novel clean cage, was not able to diminish freezing when assessed immediately after stimulation, upon stress-induced reinstatement, or spontaneous recovery. Source data are provided as a Source Data file.

gradually less prone to modulation the longer we waited to induce optical interference, and would also fit with previous findings suggesting that reconsolidation processes can be driven by mismatches in the animal's expectations in the form of prediction errors[65]. Prediction errors occur when there is a mismatch between what is expected and what happens[65]. When an organism encounters something unexpected in their environment, this can drive memory-updating processes by triggering memory destabilization[8]. Such a change in contingency promotes the disengagement of established representations in favor of novel representations[71].

Artificially reactivating a previously consolidated memory likely leads to reconsolidation of that trace as well as that of the naturally recalled fear memory. While we did not test this directly, it is this process involving plasticity that potentially confers the activated memory with the capacity to interfere with and modify the expression of the fear memory. We found this strategy was more effective at reducing fear when the competing engram was associated with a positive experience. These effects were observed in real-time during recall. We observed them persist into extinction, and prevent stress-induced reinstatement, as well as spontaneous recovery two weeks later, demonstrating the enduring nature of our manipulation. Although our effects persisted two weeks later, one potential limitation of this study is that we did not attempt to reinstate a fear memory using immediate shock at a remote time point as this is similar to stress-induced reactivation of pathological conditioned fear in PTSD, which future studies may further delineate.

Contrastingly, reactivation of an engram associated with a negative or neutral experience was not sufficient to diminish freezing when assessed during or after stimulation, upon stress-induced reinstatement, or spontaneous recovery (Fig. 8). These results, which were not related to differences in perturbed population sizes or locomotion, corroborate our previous findings[35] and highlight the importance of valence. Of note, it is possible the engrams associated with negative experiences, especially fear-conditioning, are highly similar to the fear memory thereby providing less interference. Indeed, we observed that dDG cells processing negative memories overlapped more with the fear memory at reinstatement in comparison to the neutral memories. However, we think that these results can be most parsimoniously explained within the framework of negative and positive prediction errors. Particularly, in possessing the ability to modulate the fear memory by strengthening or weakening it respectively. During recall, mice are returned to the conditioning context in which they exhibit conditioned fear as a result of their expectation of being shocked as before. When they do not receive a shock, this contradicts their expectations thereby promoting reconsolidation. One theory is that providing them with a negative experience (e.g., artificial reactivation of a negative memory such as fear-conditioning in a different context), likely resulted in their expectations being matched, and in no prediction error occurring, and thus did not lead to reconsolidation. Or a negative prediction error may have occurred (i.e. actual was worse than expected) where natural recall of the fear memory in context A plus artificial reactivation of the fear memory from context C led to reconsolidation-based strengthening of the fear memory from context A. In contrast, when mice received artificial stimulation of a positive memory, this resulted in a positive prediction error (i.e. actual was better than expected). Essentially, when mice received artificial stimulation of a tagged memory, it fell along a continuum of valence where the more positive the experience, the greater the magnitude of the positive prediction error that ensued, and the more likely this was to induce destabilization/memory-updating processes, which correlated with weakening the fear memory and decreases in fear expression.

We included closed-loop self-stimulation of the VTA as a positive experience to obtain a quantifiable measure of positive valence associated with reward, and to probe motivational aspects of operant responding for reactivation of a positive memory. Our results revealed the inherently reinforcing nature of experiencing positive affect even when it is artificially induced. Interestingly, we also observed a suppression of fear in dDG-eYFP control mice that underwent VTA self-stimulation. While these mice did not have this experience reactivated, they still experienced a reduction in fear. In humans, there is evidence to show that trait positive affect can protect against stress influencing health outcomes[72], an effect known as the undoing hypothesis[37]. In line with this hypothesis, these results suggest hippocampal involvement in processing emotional memory may contribute to its regulation of stress responses.

Activating a randomly labeled set of neurons in the dDG was sufficient to disrupt fear memory reconsolidation. Moreover, activating 40% rather than 8% of cells yielded the most robust effects on freezing. We believe this manipulation, as opposed to reactivation of a positive memory, works similarly to other stimulation-based protocols

associated with neural plasticity such as electroconvulsive therapy (ECT), deep-brain stimulation, or trans-magnetic stimulation. While there is a paucity of literature on these treatments in the context of conditioned fear with mixed effects reported[73], we think that activating a large set of randomly labeled dDG cells may perturb the system in a way that provides a reset signal, and that this type of interference in the form of optical stimulation occurring concurrently with recall of the fear memory may provide a unique window of opportunity to leverage reconsolidation mechanisms with timing of the manipulation a key factor for treatment efficacy. This is especially promising since many of these methods are already used in humans. Importantly, while one major side effect of ECT is the non-discriminatory manner in which amnesia is induced[74], our approach yielded a highly specific effect on the fear memory itself, as our manipulation did not affect a separate spatial memory, suggesting our reconsolidation-based procedure is specific to the memory recalled at the time of stimulation.

Mechanistically, we found no evidence that the competing engram or interfering ensemble was merged with the fear memory. That is to say, the cells active during reinstatement did not contain a significant proportion of cells that were part of the ensemble we artificially activated. Only in the experiment where we determined the original fear memory was altered did we observe a high degree of overlap with both the original fear memory and the interfering ensemble. Overlap between the fear memory at reinstatement and the original fear engram was significantly reduced in experimental animals suggesting a causal relationship between reconsolidation and the disengagement of these ensembles, perhaps reflecting a cellular correlate of permanently altering the fear memory.

Hallmark features of PTSD include rumination and the ability for traumatic memories to be intrusive. Our work, aimed at modulating negative memories, points to the dDG as a potential therapeutic node with respect to artificially modulating memories to suppress fear. We underscore its importance since contextual information processed in the hippocampus may have a reduced capacity to modulate fear and safety in PTSD patients. However, studies involving circuit perturbations show that engrams are distributed within the brain[75] and that the pathway from the DG to the basolateral amygdala (BLA) is necessary not only for fear-conditioning, but also to acutely rescue stress-induced depression-related behaviors[35]. Therefore, future experiments will be aimed at searching for additional treatment targets to gain mechanistic insight, as we simultaneously examine the usefulness of these manipulations within the BLA. Moreover, we will explore the development and refinement of novel modulation strategies. Alleviating cellular, circuit-level, and behavioral abnormalities comprising memory updating impairments and maladaptive conditioned behavioral states involved in disorders such as PTSD, we believe, has promising clinical significance.

## Methods
### Animals
All experimental procedures were conducted in accordance with protocol 2018000579 approved by the Institutional Animal Care and Use Committee at Boston University. Experimental mice included 283 wild-type (WT) male c57BL/6 mice (~39 days of age; Charles River Labs, 027) weighing 20–22 g at the time of arrival. Additionally, 46 WT female c57BL/6 mice (~39 days of age; Charles River Labs) weighing 18–20 g at the time of arrival were used for female exposure (Fig. 1). Two DAT^IRES-cre knock-in breeding pairs of mice were purchased from The Jackson Laboratory (B6.SJL-Slc6a3tm1.1(cre)Bkmn/J Stock No. 006660) and used to maintain an in-house breeding colony. For genotyping, a mouse tail biopsy was performed. From this colony, 50 male transgenic mice were used for the study and were approximately 34 days old at the start of the experiment and weighed between 18 and 22 g. Mice were housed in groups of 2–5 per cage. Mice were kept in a temperature and humidity-controlled colony room (temperature:

18–23 °C, humidity: 40–60%) on a regular light cycle 12:12 h light ON/OFF and experiments were run during the light part of the cycle. Cages were changed weekly and contained huts and nesting material for enrichment. Upon arrival in the facility, all mice were placed on a 40 mg/kg DOX diet (Bio-Serv, product F4159, Lot 226766) and left undisturbed for a minimum of 3 d prior to surgery with *ad libitum* access to food and water. We originally tested our hypotheses in males because contextual fear conditioning in rodents is sexually dimorphic, where males typically show higher freezing levels than females[76–78] and wanted to ensure that our effect was large enough to warrant further experimentation. We also decided to use female exposure as a positive stimulus, and exposure to males may not necessarily be a positive experience for the females. While it is extremely unfortunate that we did not include females in this study, we agree that it is absolutely essential for enhanced scientific discovery to do so in all future studies.

### Stereotaxic surgery
Aseptic surgeries were carried out with mice mounted in a stereotaxic frame (Kopf Instruments) with the skull flat resting on a heating pad. They were anesthetized with 4% isoflurane and 70% oxygen (induction), and isoflurane was reduced to 2% thereafter (maintenance). All viral constructs and coordinates are discussed below. Viral infusions were administered via a 10 μL gas-tight Hamilton syringe attached to a micro-infusion pump (UMP3, World Precision Instruments) which occurred at a rate of 100 nL/min. The infusion needle was left in place for 2 min following each infusion to avoid liquid backflow. Following injections, optic fibers were implanted and secured with two anchor screws, and a mixture of metabond and dental cement to build a head cap. All mice received 0.2 mL physiological sterile saline (0.9%, s.c.), 0.1 mL of a 0.03 mg/mL buprenorphine solution (i.p.), and meloxicam (5 mg/kg, s.c.) at the beginning of surgery. At the end of surgery, mice were placed on a heating pad and given hydrogel in addition to *ad libitum* food and water. Mice were given an additional injection of buprenorphine (0.1 mL, 0.03 mg/mL, i.p.) 8–12 h later, and 8–12 h after that both meloxicam (5 mg/kg, s.c.) and buprenorphine (0.1 mL, 0.03 mg/mL, i.p.). Apart from cage changes and being weighed, wild-type mice were left undisturbed for a 10-d period following surgery to allow for recovery and virus expression. For transgenic mice, viral expression took 6–8 weeks.

### Viral microinjections
For experiments where we labeled the cells involved in a behavioral epoch [i.e., appetitive (positive), neutral, aversive (negative)] (Figs. 1–2, 6), mice received bilateral infusions of a viral cocktail of pAAV9-cFos-tTa (UMass Vector Core - titre: $1.5 \times 10^{13}$ GC/mL) and pAAV9-TRE-(ChR2)-eYFP (UMass Vector Core - ChR2 & eYFP titre: $1 \times 10^{13}$ GC/mL) or pAAV9-TRE-(ChR2)-mCherry (UMass Vector Core - ChR2 titre: $1.1 \times 10^{13}$ GC/mL, mCherry titre: $1 \times 10^{13}$ GC/mL) in a volume of 300 nL/side at AP: −2.2, ML: ±1.3, DV: −2.0 (relative to Bregma in mm) into the dorsal dentate gyrus (dDG) and bilateral optic fibers were implanted (AP: −2.2, ML: ±1.3, DV: −1.6, relative to Bregma). For experiments where we randomly activated cells in the dDG, mice were infused with diluted (1:5) or undiluted virus pAAV5-CAMKIIa-(hChR2-H134R)-eYFP (Addgene-ChR2 titre: $1 \times 10^{13}$ GC/mL; UNC Vector Core - eYFP titre: $3.6 \times 10^{12}$ GC/mL) (300 nL/side) bilaterally into the same coordinates as above (Figs. 4–6).

For experiments where DAT-Cre mice delivered closed loop stimulation into the left VTA (Fig. 3), (AP: −3.1, ML: −0.5, DV: −4.4) we infused 450 nL of pAAV5-Ef1a-DIO-(hChR2-E123A)-eYFP (Addgene-ChR2 & eYFP titre: $2.2 \times 10^{13}$ GC/mL)[79]. For these mice, bilateral infusions of the AAV9-cFos-tTa-(ChR2)-eYFP virus were delivered into the dDG. Due to space constraints, these were delivered at AP: −1.9, ML: ±1.18, DV: −1.8 where one optic fiber was placed over the left dDG at a 9° rostral angle (AP: −1.4, ML: −1.18, DV: −1.4), and the other optic fiber was placed straight over the right dDG (AP: −1.9, ML: +1.18, DV:

−1.4). An additional optic fiber was placed straight over the left VTA (AP: −3.1, ML: −0.5, DV: −4.1).

## Experimental design

**Genetically labeling dDG neurons involved in a behavioral epoch.** We genetically labeled neurons active during distinct behavioral epochs using an activity-dependent and inducible Tet-Off (tetracycline inducible) optogenetic system[36,38]. This system involves delivery of an adeno-associated viral (AAV) cocktail that allows for the expression of a tetracycline transactivator protein as well as a tetracycline response element, that when bound allow for the expression of a light-sensitive protein e.g., channelrhodopsin-2 (ChR2) fused to a fluorescent reporter gene e.g., enhanced yellow fluorescent protein (eYFP). The system is inducible because transcription is controlled (reversibly turned on or off) by tetracycline, or more stable derivatives of tetracycline such as doxycycline (DOX)[80], which is present in the animal's diet. To label neurons involved in a particular behavioural epoch, the DOX diet is replaced with standard lab chow (*ad libitum*) 42 h prior to labeling. The system is activity-dependent because it is driven by the c-Fos promoter, which has been widely used as a neuronal marker[81]. We labeled the cells active during putatively positive, neutral, and negative experiences in the dDG. Each experience is described in more detail below. Following behavioral tagging, mice were returned to their home cages and again placed on DOX. The next day, they were fear conditioned.

**Fear conditioning.** Behavior was performed in conditioning chambers with cameras mounted to the roof for video recording (context A). Video was fed into a computer running Freeze Frame/View software (Coulbourn Instruments, Holliston, MA); freezing was measured and defined as a bout of immobility lasting 1.25 s or longer. During the fear-conditioning session, 4 shocks were delivered at 198, 280, 360, and 440 s during the 500 s session[39,41] (2 s duration, 1.5 mA intensity). Subjects were placed in a holding tank until all cage mates were fear conditioned before being returned to their home cage.

**Recall.** The following day, mice were returned to the conditioning context for a 20 min recall session. In experiment 1, mice received optogenetic stimulation (10 ms pulse width, 473 nm, 20 Hz) of the genetically labeled ensemble with blue light during either the first or last 10 min of the session. In subsequent experiments, this was conducted only during the first 10 min. We also included a control experiment where this stimulation was run during the middle 10 min of the session. Optic fiber implants were attached to a patch cord connected to a blue laser diode controlled by automated software (Doric Neuroscience Studio version 5.3.3.14). Laser output was tested at the beginning of every experiment to ensure at least 15 mW of power was delivered at the end of the patch cord (Doric Lenses).

**Extinction training.** Over the next 2 days, mice underwent 30 min extinction training sessions in the original conditioning context. For these sessions, mice were not given optical stimulation or shock.

**Immediate shock.** On a subsequent day, the animals were given an immediate shock in a new context (context B). A single shock (2 s duration, 1.5 mA intensity) was delivered in the first 2 s of the session, so animals would not form a contextual representation of the environment and therefore not form an associative fear memory but would still experience stress[44–46]. Nevertheless, the context was distinctively different from the FC context with inserts and patterned walls, different lighting conditions, and almond extract odor present.

**Reinstatement test.** The following day, mice were returned to the conditioning context for a 10 min immediate shock-induced reinstatement test.

**Spontaneous recovery.** A subset of mice was not given immediate shock following extinction. These mice were left in their home cage for 2 weeks and then put back into the conditioning context for a 10 min recall test to assess spontaneous recovery of the conditioned freezing response.

**Dependent measures.** For all sessions freezing levels were measured using Freezeview software (Coulbourn) except during recall when they were manually scored due to the presence of the optic cables that made automated scoring not possible.

## dDG-labeled behavioral epochs

**Female exposure (positive).** The experimental male mouse was placed into a clean cage with a cage top and bedding, which was used as the interaction chamber. A female mouse (PND 30–40) was then placed into the cage, and they were allowed to interact freely for 1 h[35,39].

**Novel clean cage (neutral).** Mice were placed into an empty clean home cage with bedding for 1 h[21].

**Restraint stress (negative).** Mice were placed in a restraint tube with air holes in an empty home cage with bedding for 1 h[82].

**Acute cocaine exposure (positive).** Mice were habituated to scruffing and i.p. injections with saline during handling. Cocaine hydrochloride (Sigma) was prepared in 0.9% NaCl at a concentration of 2.5 mg/ml. On the day of behavioral tagging, animals received an i.p. injection of cocaine at a dose of 15 mg/kg[53]. Immediately following the injection mice were placed into a clean cage with bedding for 1 h.

**Home cage (neutral).** Mice remained undisturbed in their home cage during the tagging window.

**Fear conditioning (negative).** Mice were fear conditioned using the same protocol as above, however, this was conducted in a distinct environment in a different room (context C), with a larger apparatus, different lighting conditions, and cues on the chamber walls[38].

**Operant responding for closed loop VTA stimulation (positive; control mice: neutral).** Customized operant testing chambers were constructed from standard mouse fear-conditioning apparatuses (Med Associates) (context E). Custom built nose-poke holes (opening diameter: 23 mm) were built into the right and left positions of one wall, and a plastic wheel (diameter: 60 mm) on a ball bearing was fixed to the right position on the opposing wall. An Arduino Mega microcontroller was used to catalog nose pokes detected via infrared sensors (adafruit), and to detect wheel manipulation via a rotary encoder (US Digital). Custom Matlab (version R2021a) code was written to deliver closed-loop stimulation, and to record nose pokes at each port (active and inactive) as well as wheel manipulations at each moment during the session. Behavioral events were recorded every 100 msec, and the turnaround time for the laser stimulation remained with the 100 ms clock period. To receive optogenetic stimulation, the mouse was required to either initiate a nose poke in the correct port or manipulate the wheel such that it turned in excess of 0.1 rev per second (or 5 revs per min) as measured continuously over a 20 ms interval. Both the nose poke and the running wheel had a lockout period of 1 sec (or 500 ms following termination of stimulation train), and the animal was required to withdraw from the nose port before receiving a subsequent stimulation. Optogenetic stimulation (10 ms pulse width, 473 nm, 40 Hz) initiated by nose poking was delivered via blue light to the VTA in 500 ms bouts. Optogenetic stimulation (10 ms pulse width, 473 nm, 20 Hz) initiated by spinning the wheel was also delivered via blue light to the dDG in 500 ms bouts.

## Experimental procedures

**Habituation and baseline.** Dat-Cre mice were given two 45 min habituation sessions with no access to the nose ports, and only access to the wheel, whereby moving the wheel had no consequences. The first session was considered a habituation session for the mice to familiarize themselves with the apparatus and no dependent measures were taken. The second session was considered a baseline session where dependent measures related to the wheel were obtained. These included the number of wheel rotation bouts, the distance the wheel traveled, the number of seconds the wheel was rotated.

**Training.** Mice were given 4 training days, one 45 min trial per day where they had access to the wheel (no consequences) and the active and inactive nose ports. Nose pokes in the active port resulted in a 500 ms (left) VTA stimulation bouts, while nose pokes in the inactive port resulted in no stimulation. Following the 3rd training session, mice were taken off DOX, and 42 h later brought back for the 4th training session. Following the session end, mice were placed back on DOX. During these sessions we continued to obtain dependent measures related to the wheel and also measured the number of nose pokes at each port as well as the number of stimulations delivered.

**Test.** The next day, mice were placed back in the operant chamber with restricted access to the nose ports and only given access to the wheel. Wheel spins during this session resulted in 500 ms dDG stimulation bouts thereby reactivating the memory of the previous training session where VTA-ChR2 mice received self-stimulation of the VTA, and VTA-eYFP mice did not. During this session, we measured the number of wheel rotation bouts, the distance the wheel traveled, the number of seconds the wheel was rotated, and also the number of simulations produced by spinning the wheel to assess whether mice would perform an operant task for reactivation of a positive memory. The following day all mice underwent fear-conditioning.

**Genetically labeling random dDG neurons not tied to encoding of a behavioral epoch.** We genetically labeled random dDG neurons driven by the CaMKIIa promoter to allow for the expression of ChR2 fused to eYFP. For the 1:5 dilution, we used sterile saline. These cells were then activated during recall. For this system, a tagging window is not required therefore, mice were not taken off DOX. However, we fed them the same DOX diet for consistency.

**Open field.** To assess whether stimulation of a cocaine engram induced locomotor activity or had any effect on anxiety-like behavior, we tagged an acute cocaine experience (15 mg/kg, i.p.) off DOX and later placed mice in an open field arena attached to a patch cord with a camera over top. The first 5 min were considered a baseline where mice received no stimulation. In the last 5 min mice received optical stimulation in the dDG (10 ms pulse width, 473 nm, 20 Hz).

**Stimulation in a novel context.** To assess whether the artificially reactivating of a tagged memory to reconsolidate a fear memory is dependent on reactivation occurring within the FC context, we reactivated the memory in a novel clean cage (context D) 24 h after fear-conditioning. We then tested recall in the original FC context 24 h later.

**Radial arm maze.** To assess whether perturbing dDG cells during fear memory recall affects other types of memory, or if the manipulation is specific to the fear memory, we trained mice on a spatial reference memory task. Mice were bilaterally infused with AAV5-CAMKIIa-ChR2-eYFP in the dDG (300 nl/side). Following surgery and 10 days of recovery, mice were then water-restricted so they would be motivated to search for a water reward in the maze. We used a custom built eight-arm radial maze made of Plexiglas (55 mm arm width, 355.6 mm arm length, center area 152.4 mm diameter) and designed a task where the

goal arm stayed consistent throughout[83]. Mice were trained to search for this location by shaping their behavior over a series of trials (see procedure below). The maze was surrounded by four curtains with distinct distal visual cues to allow mice to navigate and locate the reward using these extra-maze cues.

Initially, mice were given one 5 min habituation trial where they had access to all 8 arms which were not baited, however after the trial, mice were placed in a clear plastic container and given 1 ml of water in a falcon tube cap. The following day they were given 4 shaping trials (5 min each), and each time the goal arm was baited with 0.25 ml of water in a falcon tube cap at the end of the arm. The maze was designed with inserts within each arm to close off arms at their entry point and at the end goal location. On the first trial, mice were given access to the goal location only (both doors in the goal arms were inserted, restricting the mice to the end of the arm). On the second trial, mice were given access to the entire goal arm, but only that arm (only the door at the entry point was inserted). On the third trial, mice were placed in the center of the maze, and all arms were open. Finally, on the last trial animals were placed at a starting point (the end of the arm directly opposite the goal arm). During each trial, the mice were allowed to drink and given extra time to look up at the cues around them. In cases where mice didn't drink all the water within 5 min, we placed them in the clear plastic container and allowed them to drink the remaining amount. However, in most cases, they drank all the water in the maze.

The training lasted between 5 and 14 days. Mice were given 4 trials/day. During these trials, there was a webcam mounted over top of the maze so we could record and score behavior. We were also able to see their behavior in real time on the other side of the room divided by a thick curtain. Dependent measures included: Latency to reach the goal location (s), arm-deviations (number of arms away from the goal arm mice first visit), number of reference errors (number of arm entries into any arm besides the goal arm), repeated reference errors or working memory errors (re-entry into an incorrect arm). To reach the training criterion, mice were required to demonstrate for two consecutive days, a latency score of under 90 s, an arm-deviation score of less than 2, less than 2 reference errors, and less than 2 repeated reference errors on trial 1 and on trials 2–4 calculated separately. Mice took approximately 8 days to reach this criterion. Once they did, the following day they were fear conditioned.

**Immunohistochemistry.** Mice were overdosed with sodium pentobarbital and perfused transcardially with (4 °C) phosphate-buffered saline (PBS) followed by 4% paraformaldehyde (PFA) in PBS. Brains were extracted and stored overnight in PFA at 4 °C and transferred to a solution of 0.01% sodium azide the next day. The solution was prepared by dissolving 5 g of 10% sodium azide (Thermo Scientific) in 50 mL of 1× PBS to create a stock solution. This solution was then diluted to a 0.01% dilution by dissolving 1 mL of the 10% stock solution in 999 mL of 1× PBS. Brains were sliced into 50 μm coronal sections with a vibratome (Leica, VT100S) and collected in cold PBS. Sections were blocked for 2 h at room temperature in 1× PBS + 2% Triton (PBS-T) and 5% normal goat serum (NGS) on a shaker. Sections were transferred to well plates containing primary antibodies made in PBS-T [1:1000 rabbit anti-c-Fos (SySy 226-003, Figs. 1–5; Abcam ab190289, Fig. 6), 1:500 rabbit anti-TH (Millipore AB152, Fig. 3), 1:1000 chicken anti-GFP (Invitrogen a10262, Figs. 1–6), or 1:1000 guinea anti-RFP (SySy 390 004 Fig. 6)] and incubated on a shaker at 4 °C for 48 h. Sections were then washed 3× (10 min) in PBS-T followed by a 2 h incubation with secondary antibodies made in PBS-T [1:200 Alexa 555 anti-rabbit (Invitrogen A21428, Figs. 1–5), 1:200 Alexa 488 anti-chicken (Invitrogen, A11039, Figs. 1–6), 1:200 Alexa 555 anti-guinea (Invitrogen, A21435, Fig. 6), 1:200 Alexa 405 anti-rabbit (Abcam ab175653, Fig. 6)]. Following three additional 10 min washes in PBS-T, sections were mounted onto micro slides (VWR International, LCC). Nuclei were

**Article**

counterstained with DAPI added to Vectashield HardSet mounting medium (Vector Laboratories, Inc). Slides were then coverslipped and put in the fridge overnight to cure. The following day the edges were sealed with clear nail-polish and the slides were stored in a slide box in the fridge until imaging.

**Fluorescent confocal image acquisition and quantification.** Images were collected from coronal sections using a fluorescent confocal microscope (Zeiss LSM 800 with airyscan) at 20× magnification and Zen Blue 2.3 software. For quantification of overlaps for animals receiving bilateral viral dDG injections, 3 z-stacks (step size 0.94 μm) were taken per hemisphere from three different slices yielding ~6 total z-stacks per animal. Data from each hemisphere was then pooled and the means for the 6 z-stacks were computed. These means were then used to obtain a group mean. For histological verification of VTA injections and to confirm the VTA neurons we labeled were indeed dopaminergic, we assessed the degree to which eYFP$^+$ cells were colocalized with TH$^+$ cells. Mice receiving unilateral viral injections in the VTA had 3 z-stacks (step size 0.94 μm) obtained from three different slices yielding 3 stacks per animal. For all images, the total number of DAPI positive (+), and eYFP$^+$ neurons were counted using Image J/ Fiji software (https://imagej.nih.gov/ij/). c-Fos$^+$ neurons were stained with either RFP (Figs. 1–5) or BFP (Fig. 6) and quantified. mCherry+ neurons were also quantified (Fig. 6). Percentage of immunoreactive (eYFP, mCherry, RFP, or BFP) neurons, including overlaps, was defined as a proportion of total DAPI-labeled cells. Chance overlap was calculated as the percentage of the first immunoreactive neuron (e.g., eYFP$^+$) multiplied by the percentage of the second immunoreactive neuron (e.g., c-Fos$^+$) over the total number of DAPI neurons.

### Statistical analyses

Calculated statistics are presented as means ± standard error of the mean (s.e.m.). To analyze differences, we used one, two, and three-way analysis of variance (ANOVA) and in cases where these are repeated measures (RM) analyses, it is stated. In some cases, we used unpaired t-tests. When appropriate, follow-up post hoc comparisons (Tukey's HSD) were conducted. All statistical tests were conducted in Graphpad Prism (version 9.2.0), were two-tailed, and assumed an alpha level of 0.05. For all figures, * = $P < 0.05$, ** = $P < 0.01$, *** = $P < 0.001$, **** = $P < 0.0001$.

### Reporting summary

Further information on research design is available in the Nature Research Reporting Summary linked to this article.

## Data availability

Raw data for all experiments are available as a supplementary source data file. Source data are provided with this paper.

## Code availability

The code for the operant experiments is provided via Github (https://github.com/bladonjay/RamirezLabCode).

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

## Acknowledgements

We thank Lauren Reynolds, Emily Doucette, Daniel Sheehan, Moriah White, Heloise Leblanc, Amy Monasterio, Ryan Senne, Siria Coello, and Kaitlyn Dorst for technical assistance. We'd also like to acknowledge the use of Biorender for components of our figures (mainly the depiction of contexts and mazes the mice were placed in). These images appear in Figs. 1a, n, 2a, i, 3a, 4a, 5a, 6a, 8a, and Supplementary Figs. 2a, 3a, and 4a. This work was supported by an NIH Early Independence Award (DP5 OD023106-01), an NIH Transformative R01 Award, a Young Investigator Grant from the Brain and Behavior Research Foundation, a Ludwig Family Foundation Grant, the McKnight Foundation Memory and Cognitive Disorders Award, and the Center for Systems Neuroscience and Neurophotonics Center at Boston University.

## Author contributions

S.L.G., J.H.B., and S.R. designed the experiments. S.L.G. collected the data for all experiments/figures. Data collection for: Fig. 1 was assisted by A.H.F., E.R., and Y.Z.; Fig. 2 by A.H.F., E.R., and C.C.; Fig. 3 by J.H.B., A.H.F., L.F.R., and M.S.; Fig. 4 by A.H.F., C.C., E.R., and L.F.R.; Figs. 5, 6 by A.H.F. and A.G.; Fig. 7 by A.H.F., L.F.R., and A.G. Supplementary Fig. 1 by E.R., Supplementary Fig. 2 by all authors, Supplementary Figs. 3, 4 by E.R., Supplementary Fig. 5 by all authors. Data analysis and figures were completed by S.L.G. The manuscript was written by S.L.G. and edited by J.H.B., E.R., A.H.F., and S.R. Final edits were made by all authors.

## Competing interests

The authors declare no competing interests.
