## [Peer review file · Nature Communications]

REVIEWER COMMENTS

Reviewer #1 (Remarks to the Author):

This is a very interesting and well-crafted manuscript describing an elegant set of experiments. The main findings of the study are that the authors demonstrate that by artificially activating dorsal dentate gyrus (dDG) positive valence memory ensembles during the retrieval of a context Pavlovian fear memory test, that the context fear memory subsequently becomes modified and weakened. Additionally the authors show that if they activate a larger, non-specific population of neurons within the dDG during context fear memory recall, the fear memory subsequently becomes modified and weakened. This is a beautiful body of work that will be highly influential, not only for the learning and memory field but also for potential therapeutic avenues to weaken pathological fear memory. I do however have a few major concerns regarding the conceptual interpretation the authors make regarding the interpretation of the data and potential experimental controls that would have been nice to include in this study.

1. My biggest issue is how the authors describe and interpret the reduction in fear memory strength as a product of “reconsolidation-interference”. It is possible that reconsolidation processes are being interfered with but this is difficult to demonstrate and there is no evidence presented in the manuscript that reconsolidation is being interfered with per se. Rather I would state that the fear memory is being updated/modified via reconsolidation when the positive valence memory is activated during fear memory recall and this modification leads to fear memory weakening. Historically reconsolidation has been viewed as the molecular and cellular process that occurs to re-stabilize a retrieved memory that has initiated another bout of plasticity induced by memory retrieval. This destabilization requires new RNA and protein synthesis to re-stabilize the memory trace. In light of this, when gene expression/protein synthesis have been inhibited either by gene knockdown or RNA or protein synthesis inhibitors, the molecular and cellular processes of reconsolidation are believed to be “interfered” with. I don’t believe we can assume that just because a memory is weakened, it must be due to interfering with the reconsolidation of that memory. It seems more likely that sometimes the weakening of the memory is the result of the reconsolidation process. After all reconsolidation is believed to be the process where an existing memory can be modified – either strengthening or weakening the memory. I think our interpretation of how and why the memory is weakened would determine if we interpret this as “interference” of reconsolidation or just the natural process of reconsolidation updating the memory. In light of no additional evidence or mechanisms presented or discussed, I would suggest not to describe this memory weakening phenomena as interference of reconsolidation.

2. I think the authors need to perform an experiment where they activate the positive valence engram OR the non-specific labeled dDG neurons with AND without fear memory recall to determine if the fear memory modification depends on the activation of the fear memory. This is a hallmark reconsolidation experiment. And I would not assume that activation of the positive valence engram AND activation of non-specific dDG both lead to fear memory weakening in a retrieval dependent memory.

3. I found figure 5 very difficult to understand. It might help to present the data in the order they are collected rather than grouping the fear data separately from the maze data. Also the results section dedicated to this section could be improved for clarity.

4. Regarding figure 5. I would like to know, if you activate a lot of non-specific dDG neurons during recall of the maze task, can you modify the maze memory. However it does not appear that was the experiment that was conducted. It appears what was done was to determine if activating dDG neurons during fear memory recall would also modify a non-recalled maze memory.

5. Considering the focus of the manuscript is on reconsolidation, I would like the authors to show us the freezing data during the first ~3 minutes of context exposure on day one of extinction separately (for every applicable experiment). This would be similar to Post-Reactivation LTM test (PR-LTM). If the memory modification is strong, we should see a difference in freezing there. However if it is not, maybe we will only see it at the reinstatement and spontaneous recovery tests. Regardless of the outcome this would provide important information that could guide the reconsolidation field. Notably, collapsing the extinction data,(as the authors have done), could easily mask subtle differences in fear memory strength that may only be observable early during this PR-LTM like time period.

6. For experiments where the authors randomly activated cells in the dDG, using a diluted AAV (1:5000). Are you sure this 1:5000 number is correct? If you start with $1E13$ GC/mL. 5000 fold would be $2E9$ GC/mL. I would be surprised you would see any viral transduction at that dose.

Reviewer #2 (Remarks to the Author):

This study can be thought of as an intriguing marriage of two different research topics and research strategies: so-called 'engram' research, and research focussing on the phenomenon of 'reconsolidation'. Engram research generally involves inserting a memory into a mouse (sometimes referred to as 'Inception'), whereas in a typical reconsolidation experiment, a memory is activated, rendering it labile, at which time the memory can be disrupted or perhaps updated, usually by pharmacological means. The present study combines these approaches by attempting to disrupt a reactivated memory by inserting a "competing" memory. Conceptually this approach seems to have exciting potential.

I have taken a first stab at reviewing this dense and multi-experimental paper, but I feel I will need to look at it a second time. Generally speaking, I think the paper could have much greater impact with more exposition that will help the general reader understand the rationale, design, and implications of the results.

For example, the rationale for the stages of testing – e.g., in Expt 1 Fear conditioning->Recall->Extinction->Immediate shock->Reinstatement – is not well spelled out for the general reader. The reason for each of the stages/tests in each experiment, what results might be expected and what those results would mean is not well explained. For example, which of these stages is the critical test, and which represent reactivation of the memory? This may seem obvious to the authors who are very close to the work, but it could be difficult for the non-expert. If 'Recall' is the reactivation stage, during which the reactivated labile memory is subjected to disruption by the inserted memory, then make that clear. If the critical test is Reinstatement, then why are measures taken during reactivation, and what are the implications of those measures? More of these 'why?' and 'what does it mean?' questions need to be answered explicitly for the general reader, at every stage of every experiment.

Such additional 'spelling out' will help to answer many questions the general reader may have. For example:

Some experiments (e.g., Expt 1) test effects on reconsolidation of activating different types of memory. The comparison is between Positive, Neutral, and Negative memories. This being science, we want all those conditions to be exactly the same with the exception of the independent variable of interest, Valence (Positive, Neutral, or Negative). However one can see at a glance that the 3 conditions differ in many other ways. That means confounds. For example, the physiological state of the animal will be very different across the conditions, and the valences are more or less similar to the (negative) valence of the fear test (as mentioned by the authors in the Discussion). Could the experiment not have been conducted in a less confounded manner? The authors may have an answer to this, but this is the sort of thing that needs to be made explicit.

The previously tagged DG ensembles (memories) were activated either in the first or second half of the session. Why? Later in the paper the authors say it is to reduce the amount of stimulation. Why? Is this because the brain heats up during light stimulation (it does)? Then is the full length of a session the amount of light it takes to heat up the brain? Is half a session now short enough that the authors can be convinced this is not a confound? Again, spell it out. But even if it makes sense to limit the amount of light, we still don't have any rationale for the choice of early versus late activation. Indeed the discussion of this experiment (in the Discussion section) suggests that the late activation may have introduced unwanted (?) factors that hinder interpretation of the results. It seems their explanation is testable (by leaving the animal in the chamber, I guess), so why did they not test it? The two conditions (early and late) are certainly different in that the early allows examination of post-stimulation freezing during the session, whereas the late stimulation does not.

There is not a “no light” condition, as is typical in optogenetic experiments. Is this not needed for these experiments. If not, Why?

During activation of the interfering memory, the behaviour of the animal can change drastically depending on the condition. This introduces a confound that could affect the results when the animal is tested later. Can the authors explain why readers should not be concerned about this confound?

Some data are a just a bit sketchy. For example, in Figure 1C, the Negative eYFP condition, there is a bimodal distribution such that 2 points are very high, and 3 points are very low. These animals are clearly very different, and in fact that difference is probably the largest numerical difference reported in the whole study. I would have redone this condition/experiment, but at the very least again, the authors should explain why we should trust data like this. Do these data even conform to the minimum requirements for the statistics?

What is the rationale for using only male mice?

Viruses in DG are known to kill adult born neurons. The authors should explain why this is not a problem in this study.

“Activation of randomly labelled dDG neurons is sufficient to disrupt fear consolidation.”
“Reconsolidation-interference can be effectively achieved by activating ensembles that are not connected to engram of a particular valence if enough cells are activated.” The general, reader may conclude that the experiments using memory insertion as disruption are meaningless, as any random messing around in the DG will yield the same effect, which is not unexpected and not very interesting. The authors need to do a better job of clarifying this for the reader.

If the authors can provide a bit more speculation about explanation of mechanism – why, for example, different valences of interfering memories have different effects – that could help make the discussion a little more satisfying than it is currently.

The interfering-memory approach is suggested as a target for therapy for PTSD. Many readers will know that optogenetics is not feasible in humans. So how exactly will this work translate? Opto in the future? Drugs that target the DG? What’s wrong with drugs that have been used previously, e.g., propranolol? Does this kind of approach bring any advantage above and beyond propranolol? More of this in the Discussion in necessary especially since the paper is set up to be all about PTSD.

More minor comments

Reference to data figures in the Results section should be made in sentences referring to the data, not to restatements of the methods.

Check rules for hyphenation.

Reviewer #3 (Remarks to the Author):

Summary

Grella et al., test the hypothesis that positively-valenced memories (exposure of a female conspecific, delivery of cocaine) formed in a novel environment can interfere with the reconsolidation of an aversive memory (e.g. a conditioned contextual fear memory). The authors approach this goal by combining the Tet-tag system, optogenetics, extensive behavioral testing, and immunohistochemistry in the dorsal dentate gyrus (dDG). The overarching question posed by this research – can rewarding experiences counteract negative experiences or even “re-write” them? – is both important and timely. The manuscript provides a plethora of interesting data. However, a number of important concerns make it difficult to support all of the conclusions drawn by the authors. These are highlighted below. Most warrant significant revisions to the text, not necessarily new experiments.

Concerns

- 1) The single most important issue is the question of whether or not the authors believe that the engram for a contextual fear memory is indeed stored in DG. There is an entire literature pointing to fear memory engrams being stored in the amygdala, with engrams for spatial/contextual information being stored in the hippocampus (e.g. starting with Kim and Fanselow, 1992, to Rashid et al., 2016, and numerous more). In other words, one common view is that the hippocampus encodes contextual information, which then gets sent to the amygdala for formation of a contextual fear memory engram. These data (as well as previous data published by this research group), seem to contradict this. While this reviewer is not suggesting more experiments to address this discrepancy, I believe that a section on this matter in the discussion is highly warranted.
- 2) The authors link their results to relevance for PTSD, indeed they state in the intro that “pathological conditioned fear can occur for decades even in the absence of the exact context in which the traumatic event took place.” Given that they are looking at contextual fear in particular here, it seems that their findings are more relevant to specific phobias rather than PTSD. Related to this, in the second paragraph

of the intro, the authors link their findings to “maladaptive fear”, however, again, they are not modeling a maladaptive fear here, but are rather examining an adaptive, conditioned context fear. Text should be adjusted to reflect this.

3) Supp figure 1 is a lot of data points with no information as to which data point belongs to which animal in which group. There are a few data points showing pre-conditioning freezing (!). Which animals were these? Please re-graph this to include group identities.

4) The authors spend a lot of time highlighting reconsolidation. They present data on reinstatement and spontaneous recovery as indirect read outs of whether or not reconsolidation has been interrupted. However, their extinction sessions (in particular the first one), are also powerful read outs. But because this data is being collapsed across freezing during the entire extinction session, the authors may be missing out on important data. The authors should instead break down the extinction sessions so that freezing during the first few minutes (and the effects of their manipulations on these time points) can be investigated. This is especially important as the authors repeatedly make claims about extinction “rate” (e.g. Fig 1j), but never present any data or statistical analyses looking at/comparing the actual rate (i.e. extinction curve). This data can be put in the supplemental.

5) Fear acquisition curves (instead of bar graphs) should be shown. This will allow the authors to determine whether any of their manipulations affect the rate of fear acquisition.

6) The first experiment reveals that there are interfering effects with “neutral” experiences. The authors then shift to a homecage exposure in experiment 2, claiming that the original “neutral experience” in a novel clean cage may actually be positive. If anything, exposure to a novel environment might produce some neophobia, however the authors do not address this.

7) Related to point 6. The “homecage “neutral” environment represents an environment that the animal has been in for an extended period of time. Such a condition is often used as a homecage control wherein almost no cfos expression is present in the dorsal hippocampus (e.g. work from Barnes lab). How are the authors “tagging” activated neurons in this condition if there are usually almost no cfos+ cells in this homecage condition??

8) It is interesting that memory “interference” is only effective when the authors activate positively-valenced, tagged ensembles at the beginning of the reactivation test. However, their explanation that this is because the second half of the reactivation session ends with transport back to the homecage remains untested. In order to test this, the authors should run a control experiment where activation occurs in the middle of the 20 minute reactivation session (eg min 5-15), and examine whether interference is still intact. Otherwise, they cannot rule out that the reason the first 10 min activation works better is because animals haven’t begun to extinguish their fear.

9) In the discussion, the authors bring up the interesting possibility that there may be less interference of a contextual fear memory by an aversive experience because there is more overlap of engrams. This would be consistent with the data that the authors show regarding activation of a random subset of DG neurons and the ability of this activation to similarly interfere with contextual fear memory reconsolidation. Hence, this is the most parsimonious explanation for the data. For this reason, the real finding of the paper is that fear memory reconsolidation can be perturbed if non-overlapping cell

populations are activated in DG during retrieval of that fear memory, rather than anything specific about a positively-valenced experience.

Minor issues

- 1) In the intro (first paragraph) the authors state that footshocks elicit freezing, they don't (they elicit activity bursts). Only the CS (in this case a context) elicit freezing (a CR) via conditioning.
- 2) Methods. Were mice run during the light or dark part of the cycle?

REVIEWER COMMENTS

Reviewer #1 (Remarks to the Author):

This is a very interesting and well-crafted manuscript describing an elegant set of experiments. The main findings of the study are that the authors demonstrate that by artificially activating dorsal dentate gyrus (dDG) positive valence memory ensembles during the retrieval of a context Pavlovian fear memory test, that the context fear memory subsequently becomes modified and weakened. Additionally, the authors show that if they activate a larger, non-specific population of neurons within the dDG during context fear memory recall, the fear memory subsequently becomes modified and weakened. This is a beautiful body of work that will be highly influential, not only for the learning and memory field but also for potential therapeutic avenues to weaken pathological fear memory. I do however have a few major concerns regarding the conceptual interpretation the authors make regarding the interpretation of the data and potential experimental controls that would have been nice to include in this study.

We thank the reviewer for these laudatory comments. We have done our best to address each of the reviewer's concerns below, and all changes to the manuscript appear in red.

1. My biggest issue is how the authors describe and interpret the reduction in fear memory strength as a product of "reconsolidation-interference". It is possible that reconsolidation processes are being interfered with, but this is difficult to demonstrate and there is no evidence presented in the manuscript that reconsolidation is being interfered with per se. Rather I would state that the fear memory is being updated/modified via reconsolidation when the positive valence memory is activated during fear memory recall and this modification leads to fear memory weakening. Historically reconsolidation has been viewed as the molecular and cellular process that occurs to re-stabilize a retrieved memory that has initiated another bout of plasticity induced by memory retrieval. This destabilization requires new RNA and protein synthesis to re-stabilize the memory trace. In light of this, when gene expression/protein synthesis have been inhibited either by gene knockdown or RNA or protein synthesis inhibitors, the molecular and cellular processes of reconsolidation are believed to be "interfered" with. I don't believe we can assume that just because a memory is weakened, it must be due to interfering with the reconsolidation of that memory. It seems more likely that sometimes the weakening of the memory is the result of the reconsolidation process. After all reconsolidation is believed to be the process where an existing memory can be modified – either strengthening or weakening the memory. I think our interpretation of how and why the memory is weakened would determine if we interpret this as "interference" of reconsolidation or just the natural process of reconsolidation updating the memory. In light of no additional evidence or mechanisms presented or discussed, I would suggest not to describe this memory weakening phenomena as interference of reconsolidation.

We completely agree with the reviewer that it makes more sense to present these findings as a modification (i.e. weakening) of the fear memory through reconsolidation where the fear memory is being updated through incorporation of the positively-valenced information presented during recall through artificial stimulation. This is especially so since, as the reviewer states, we have not specifically "interfered" with any molecular processes (e.g., using protein synthesis inhibitors) and are likely promoting reconsolidation by creating a prediction error (see below) rather than interfering with it. Indeed, we agree with the review that the weakening of the fear memory is the result of our manipulation during the reconsolidation process, not the interference of these processes. We have amended the title of the paper, the abstract, results, and discussion to reflect this.

We have also rewritten the last paragraph of the intro which referred to our manipulation as "interference" and removed this description, replacing it with the phrase "competing engram" to be consistent with our

discussion. We have now used language that clearly states that our manipulation occurs during the reconsolidation window (i.e. to take advantage of this process rather than interfere with it) throughout the manuscript. In most cases, we replaced “reconsolidation-interference manipulation” with “reconsolidation-based manipulation” and only used the interference to describe the concept of two memories competing within the hippocampus and not to refer to reconsolidation processes.

2. I think the authors need to perform an experiment where they activate the positive valence engram OR the non-specific labeled dDG neurons with AND without fear memory recall to determine if the fear memory modification depends on the activation of the fear memory. This is a hallmark reconsolidation experiment. And I would not assume that activation of the positive valence engram AND activation of non-specific dDG both lead to fear memory weakening in a retrieval dependent memory.

We thank the reviewer for this suggestion and agree it is a necessary control experiment to include. We have conducted this experiment using the Tet-Tag system to label a positive (i.e. female exposure) experience and then artificially reactivated this tagged engram in a context other than the fear-conditioning context (clean cage) 24 hrs prior to the recall test. These methods have been added to the methods section. The results have been added to the results section and are depicted in the new Supplemental Figure 4. We appreciate this experiment having been suggested since this figure not only shows the important result that positive memory reactivation, as a means to modify a fear memory, must occur during reconsolidation of that fear memory (i.e. naturally prompted here by the conditioning context), but it also provides a nice, direct visual comparison to the other positive memory groups with respect to the timing of the optical stimulation, and how that timing was critical to the outcome. We have also added these results to the discussion in its first paragraph.

3. I found figure 5 very difficult to understand. It might help to present the data in the order they are collected rather than grouping the fear data separately from the maze data. Also, the results section dedicated to this section could be improved for clarity.

We thank the reviewer for this comment and have taken the comment into careful consideration throughout figure 5. Given the way the data were analyzed we think it makes the most sense to keep the maze data together and the freezing data together. However, we have substantially amended this figure for clarity. We have revamped the experimental design portion of the figure to be less busy, using a colored backdrop to separate it from the data itself. We have also included icons of the maze when the mice were tested on spatial ability. These are now color-coded to match the data presented below. For example, the fear conditioning component of the experimental design schematic has a purple maze next to it, and below where that data is illustrated, there is a new label that says fear conditioning (rather than just FC) and it is written in the same color. We have done this for each time performance on the maze was tested. We have also added the word acquisition under the acquisition graphs and changed “recall” to “fear memory recall” to make a clearer connection between the experimental design schematic and the graphs below. The data are presented in the order they were collected, and you can now see exactly which days we are comparing to which. Moreover, we have rearranged the order of the graph to make more sense (e.g., the days to reach criterion is now with the maze data) and overall, tried to polish up the entire figure including making the criterion line stick out less. All of the IS-RE comparisons (for every figure including this one) have been redone to show post-hoc comparisons on the graph. We have added titles in the figure that specify what the implication of the data sets are. Section 1 is the experimental design schematic, bordered off; Section 2 is the “Effect of DG Stimulation (during fear memory recall) on Fear Memory”, while Section 3 is the “Effect of DG Stimulation (during fear memory recall) on Spatial Reference Memory”. We have also rewritten the figure caption and the results section of this experiment with the intention of significantly improving clarity and readability.

4. Regarding figure 5. I would like to know, if you activate a lot of non-specific dDG neurons during recall of the maze task, can you modify the maze memory. However, it does not appear that was the experiment that was conducted. It appears what was done was to determine if activating dDG neurons during fear memory recall would also modify a non-recalled maze memory.

We thank the reviewer for this comment. We specifically ran this experiment as a control to ensure the memory that was affected by the optical stimulation manipulation was solely the fear memory (due to reconsolidation of the fear memory) and that this effect didn't spread to other types of information or memory. From a therapeutic standpoint, we felt it was necessary to show this specificity, especially given that other stimulation treatments such as electroconvulsive therapy (ECT) have reported memory deficits on a more general level such as retrograde amnesia ranging from 30-55%. While it would be interesting to assess whether this reconsolidation-based optical stimulation protocol could interfere with a spatial reference memory, we believe this is beyond the scope of the current study; and, its feasibility for the purposes of this paper is difficult given how many months it took to run this experiment. However, we do think this is a very interesting research question and it is something we intend to investigate further in future studies. Indeed, we will be conducting studies relating to other types of memory using this procedure, including an ongoing study in the lab manipulating drug-related memories using similar methods.

5. Considering the focus of the manuscript is on reconsolidation, I would like the authors to show us the freezing data during the first ~3 minutes of context exposure on day one of extinction separately (for every applicable experiment). This would be similar to Post-Reactivation LTM test (PR-LTM). If the memory modification is strong, we should see a difference in freezing there. However, if it is not, maybe we will only see it at the reinstatement and spontaneous recovery tests. Regardless of the outcome this would provide important information that could guide the reconsolidation field. Notably, collapsing the extinction data, (as the authors have done), could easily mask subtle differences in fear memory strength that may only be observable early during this PR-LTM like time period.

We thank the reviewer for this extremely useful suggestion. We have reorganized the data and included a new Supplemental Figure 1, which shows the fear acquisition curves and the extinction data for the first 3 minutes of the first context exposure post recall, and the last 3 min of extinction training on day 2. We performed this for all experiments. Additionally, we ran several new control experiments that were included in the other supplemental figures - for these figures we have included both the acquisition curves and the extinction data showing the first and last 3 min as well. The results have been incorporated into the results section and the statistics included in the figure captions. The implications of these results have been incorporated into the discussion. Generally, our data suggest a stronger effect during the first 3 min of extinction when we activate a random group of DG neurons with the CAMKIIa promoter virus rather than using the c-Fos-tTA virus to tag differently valenced experiences. However, when we look at the positive memory reactivation in comparison to the the No Light control groups (New Supplemental Figure 3) or simply on its own in comparison to eYFP control groups at either the first 10 min, the last 10 min or the middle 10 min (New Supplemental Figure 2), we do see significant differences in the first 3 min of extinction. Thus, overall our data suggest that memory modulation is, in certain circumstances, strong enough to see a group difference at this time point.

More specifically, we found that for all experiments there was a main effect of time. For experiment 1 (Figure 1, New Supplemental Fig 1a-c), there was a significant main effect of virus for mice that were stimulated in either the first or last half of the recall session, but post-hocs revealed no significant group differences during the first 3 min of extinction. For the mice in the spontaneous recovery experiment, we

saw no main effect of virus, only time. For experiment 2 (figure 2, New Supplemental Fig 1d), there was a significant virus x time interaction but again no group differences in the first 3 min of extinction. For the VTA experiment (Figure 3, New Supplemental Fig 1e), there was significant main effect of VTA virus and a significant Time x DG Virus interaction and a significant group difference between the VTA Chr2 and VTA-eYFP groups in the last 3 min of extinction and between the VTA-CHR2-dDG-ChR2 group and the VTA-CHR2-dDG-eYFP group but no group differences in the first 3 min of extinction. We then noted differences in the first 3 min of extinction when we looked at stimulating the non-engram DG cells with the CAMKIIa promoter virus (Figure 4). There was again a significant effect of time, but also virus and a significant difference between the diluted Chr2 group and diluted eYFP group (New Supplemental Fig 1f). For the maze experiment (Figure 5), where we used this same viral strategy, there was a significant time x virus interaction with general group differences between the Chr2 and eYFP groups despite reminder status (New Supplemental Fig 1g). And for experiment 6 (Figure 6, New Supplemental Fig 1h) we see a main effect of time and a distinct group difference at both the first and last 3 minutes of extinction. When we compare the Chr2 and eYFP groups that received stimulation in the first half of the recall session (Figure 1) with the no light groups (New Supplemental Fig 3), we do then see a significant Light x Virus interaction with a group difference observed. Finally, when we compared time of stimulation: either first, middle, or last half of the session (New Supplemental Fig 2) we found a significant time x virus interaction with a group difference only in the group that got stimulation in the first 10 min.

6. For experiments where the authors randomly activated cells in the dDG, using a diluted AAV (1:5000). Are you sure this 1:5000 number is correct? If you start with 1E13 GC/mL. 5000-fold would be 2E9 GC/mL. I would be surprised you would see any viral transduction at that dose.

We thank the reviewers for this comment and for noticing this error. We checked our notes from the dilution and where we had prepared a dilution of 1ul of virus in 5ul of sterile saline it was written 1ul of virus in 5ml of sterile saline. We have changed the methods, results, figure captions and figures to reflect this correction. We had originally decided to use this dilution since they used a similar dilution in Jimenez et al., (2020).

We have included representative images of this dilution in the paper Fig. 7c and Supplemental Fig. 5e. We have also provided additional images here.

Reviewer #2 (Remarks to the Author):

This study can be thought of as an intriguing marriage of two different research topics and research strategies: so-called ‘engram’ research, and research focussing on the phenomenon of ‘reconsolidation’.

Engram research generally involves inserting a memory into a mouse (sometimes referred to as 'Inception'), whereas in a typical reconsolidation experiment, a memory is activated, rendering it labile, at which time the memory can be disrupted or perhaps updated, usually by pharmacological means. The present study combines these approaches by attempting to disrupt a reactivated memory by inserting a "competing" memory. Conceptually this approach seems to have exciting potential.

I have taken a first stab at reviewing this dense and multi-experimental paper, but I feel I will need to look at it a second time. Generally speaking, I think the paper could have much greater impact with more exposition that will help the general reader understand the rationale, design, and implications of the results.

We thank the reviewer for these laudatory comments. We have done our best to address each of the reviewer's concerns. All manuscript changes appear in red.

1. For example, the rationale for the stages of testing – e.g., in Expt 1 Fear conditioning->Recall->Extinction->Immediate shock->Reinstatement – is not well spelled out for the general reader. The reason for each of the stages/tests in each experiment, what results might be expected and what those results would mean is not well explained. For example, which of these stages is the critical test, and which represent reactivation of the memory? This may seem obvious to the authors who are very close to the work, but it could be difficult for the non-expert. If 'Recall' is the reactivation stage, during which the reactivated labile memory is subjected to disruption by the inserted memory, then make that clear. If the critical test is Reinstatement, then why are measures taken during reactivation, and what are the implications of those measures? More of these 'why?' and 'what does it mean?' questions need to be answered explicitly for the general reader, at every stage of every experiment.

We thank the reviewer for this comment and agree that we should give more detail in the main text rather than just the methods regarding the rationale for our experimental design. We have restructured the entire first paragraph of the results section to make it explicitly clear why we chose the time intervals we did, the sessions we did, and what we were expecting to occur at each stage. We think this greatly improves the readability of the paper.

We explicitly chose to use the reinstatement model where mice are conditioned, go through extinction training to extinguish this response, and then undergo exposure to a stressor to reinstate the conditioned response. We used a strong 4 shock FC protocol given that we wanted to model aspects of PTSD and this protocol was taken from two papers (Constanzi et al., 2014; Redondo et al., 2014) and was also used in Doucette et al., (2020). We have added this third paper to the references. We used a 20 min recall test so that our manipulation would occur during the reconsolidation period of the fear memory rather than a longer session where extinction learning might begin to take place (Suzuki et al., 2004). We have added this reference to the paper. We simultaneously stimulated the tagged dDG ensembles during only the first or last half of the session rather than the entire session, as we did not want to risk damage to the brain by heat (Arias-Gil et al., 2016) and wanted to compare light on and light off periods. We initially hypothesized that reactivating a positive memory during the last half of the recall session would promote reconsolidation because the fear memory would be online with the potential of being altered / updated to reflect decreased fear expression when fear was assessed at subsequent time points. Here we were interested in whether we would see real-time decreases in freezing concurrent with the optical stimulation during recall. But more importantly, we wanted to measure the potential long-lasting effects of our manipulation and thus extended our assessment to include tests for stress- induced reinstatement after an immediate shock in context B or spontaneous recovery. For immediate shock we used a clever procedure developed by Fanselow (1986; 1990) (Fanselow et al., 1994) where the shock is delivered in a

new context in under 2 s so that the mice do not form a contextual representation of the environment and therefore are not reconditioned to form a new associative fear memory but still experience stress (almost in the absence of space) that closer represents the type stress reactivity you might see in PTSD with overactive brain stress systems characteristically present. We have added these references to the paper.

Such additional 'spelling out' will help to answer many questions the general reader may have. For example:

2. Some experiments (e.g., Expt 1) test effects on reconsolidation of activating different types of memory. The comparison is between Positive, Neutral, and Negative memories. This being science, we want all those conditions to be exactly the same with the exception of the independent variable of interest, Valence (Positive, Neutral, or Negative). However, one can see at a glance that the 3 conditions differ in many other ways. That means confounds. For example, the physiological state of the animal will be very different across the conditions, and the valences are more or less similar to the (negative) valence of the fear test (as mentioned by the authors in the Discussion). Could the experiment not have been conducted in a less confounded manner? The authors may have an answer to this, but this is the sort of thing that needs to be made explicit.

We thank the reviewer for this comment which indeed has helped clarify many experimental descriptions in our manuscript. As mentioned, the independent variable of interest is Valence. With this in mind, we did our best to control for all other factors. For example, in exp 1 (fig 1), we controlled for the length of time each mouse received each behavioral epoch (1 hr) as well as the environment in which they received this experience (novel clean cage). We chose these specific experiences to tag since they were used in our previous studies. One of the main reasons we used female exposure as the initial positive experience in this study was because this study was a follow up to Ramirez et al., (2015) where we tagged a similar experience and found that artificial reactivation of this memory was able to rescue stress-induced behavioral impairments. We have now added this rationale and citation to the introduction. Similarly, we tagged a single bout of immobilization stress as the starting negative experience in this study since that was the same negative experience used in the Ramirez et al., (2015) study. In the methods section for each type of tagged experience used, the associated citation can be found.

For exp 3 (fig 3) we also controlled for the length of time each mouse received each behavioral epoch where all animals were placed in the training environment for 45 min per day, hooked up to the optical cords the same way, and allowed to interact with their surroundings in the same manner across groups. The only difference was the virus they were injected with and thus the consequence of receiving optical stimulation.

For exp 2 (fig 2) this was slightly more challenging since the negative experience was fear-conditioning, the neutral experience was remaining in the home cage, and the positive experience was an injection of cocaine. However, in Ramirez et al. (2015) we also tagged a home cage experience and in Liu et al., (2012) we tagged a fear conditioning experience and we didn't want to deviate from the experiences used in our previous studies.

We also conducted a serological assessment to ensure that the size of the tagged engrams and the number of the DAPI labeled cells were similar across groups. The remainder of the experiments did not include a tagged memory component (Figs 4-6) or did not compare differently valenced tagged experiences (See supplemental figs). In terms of the physiological state of the animal, we believe this is an aspect of the experience itself and therefore we did not view this as a confound, but rather the intended manipulation of the independent variable Valence.

3. The previously tagged DG ensembles (memories) were activated either in the first or second half of the session. Why? Later in the paper the authors say it is to reduce the amount of stimulation. Why? Is this because the brain heats up during light stimulation (it does)? Then is the full length of a session the amount of light it takes to heat up the brain? Is half a session now short enough that the authors can be convinced this is not a confound? Again, spell it out. But even if it makes sense to limit the amount of light, we still don't have any rationale for the choice of early versus late activation. Indeed, the discussion of this experiment (in the Discussion section) suggests that the late activation may have introduced unwanted (?) factors that hinder interpretation of the results. It seems their explanation is testable (by leaving the animal in the chamber, I guess), so why did they not test it? The two conditions (early and late) are certainly different in that the early allows examination of post-stimulation freezing during the session, whereas the late stimulation does not.

We thank the reviewer for this comment. We have restructured the first paragraph of the results to provide the rationale for the time intervals we used. We also included this information for the rest of the experimental design and our expected findings. We hope this improves the readability of the paper.

We specifically used a 20 min recall test so that our manipulation would occur during the reconsolidation period of the fear memory rather than a longer session where extinction learning might begin to take place (Suzuki et al., 2004). We stimulated the tagged dDG ensembles during the first or last half of the session rather than the entire session, as we did not want to risk damage to the brain via heat due to even more extended periods of optical delivery (Arias-Gil et al., 2016) which is possible as the reviewer mentioned. Experimentally, we also wanted to compare light on and light off periods for within-subject analyses. We initially hypothesized that reactivating a positive memory during the last half of the recall session would promote reconsolidation and that conducting the stimulation in the first half would be a weaker manipulation because the memory would not have been already online. We didn't know what to expect with the stimulation occurring in the first ten min but included it as a counterbalanced control of the other condition. We did not anticipate that the time in the chamber post-stimulation could potentially serve as a substrate for learning. As a control, we have included a group that received stimulation in the middle of the session and compared these findings to the other two stimulation intervals. This design ensures the session length, and the stimulation length are kept constant. In this way we are able to see pre-stimulation freezing and post-stimulation freezing. We did not run the experiment this way initially because we wanted to compare the manipulation when it occurred post-memory reactivation and when it did not and reasoned that having all three groups would yield a more robust data set to compare between. These new results are included in the results section and the analyses in the figure caption for New Supplemental Fig. 2. The methods have been added to the methods section and the implications are further discussed in the results and discussion. Specifically, we found that the effect was similar in stimulation in the first ten min in most respects but was slightly less effective depending on which behavioral test was subsequently measured. Accordingly, we have reframed the discussion to include a few sentences about the potential role that "expectations" may be playing in our behaviors. The mice are expecting to get shocked based on the previous day, we speculate the positive prediction error induces a mismatch of their expectations and that this promotes reconsolidation. However, we believe the early stim protocol works best because the mice were shocked between minute 3-7 the prior day and the early stim protocols (0-10 min and even the 5-15 min) occur during this time period. We therefore posit that the manipulation works best when it induces the highest mismatch in expectations.

While we have not done specific experiments to compare different time periods of optical stimulation to see how it affects cellular physiology, we have used a 10 min interval previously and have not seen any notable negative outcomes. For instance, the animals do not behave differently (e.g., eYFP controls),

exhibit any indications of distress or pain, and under the microscope the cells look healthy. Given that we had decided on a 20-min recall session, and we had used a 10-min stimulation period previously (Ramirez et al., 2015; Chen et al., 2019; Doucette et al., 2020) we settled on this procedure.

4. There is not a “no light” condition, as is typical in optogenetic experiments. Is this not needed for these experiments? If not, Why?

We thank the reviewer for this suggestion. We have included two additional groups to address this. Depicted in the new Supplemental Figure 3 is a comparison of 4 groups. All groups were given a 1h positive experience (female exposure). Two of the groups were injected with ChR2 (c-Fos-tTA-TRE-ChR2-eyFP) and of the other two groups, one was given eYFP (c-Fos-tTA-TRE-eyFP) and the other No Virus. This was done to control for the light as well as the presence of the virus itself. Mice were fear-conditioned, and then during the first half of the recall session, half the ChR2 mice were given optical stimulation (Laser On), and the other half were given no light (Laser Off). The eYFP group was also given light while the No Virus group was given no light. The groups were:

Two existing groups:

- 1.ChR2 - Laser On (Fig 1h-m)
- 2.eYFP - Laser On (Fig 1h-m)

Two new groups:

- 3.ChR2 - Laser Off
4. No Virus - Laser Off

We found that only mice that received both optical stimulation and the ChR2 virus exhibited a decrease in freezing during the recall session, during the first day of extinction (entire session and during the first 3 min), as well as during reinstatement, and when comparing immediate shock to reinstatement. This demonstrates that the ChR2 virus on its own is not responsible for the effect, nor is the light on its own in the absence of the ChR2 protein. This also gives us a true baseline of the behavior without any viral or optical manipulation at all to put the findings into context. These results are now included in the results section and the discussion (first para) and the stats are in the figure caption and the methods added to the methods section.

5. During activation of the interfering memory, the behaviour of the animal can change drastically depending on the condition. This introduces a confound that could affect the results when the animal is tested later. Can the authors explain why readers should not be concerned about this confound?

We thank the reviewer for this question. In a separate ongoing study (unpublished), we have demonstrated that tagging a positive cocaine-related memory does not induce any locomotor effects when reactivated in a novel context. In the current study, we specifically ran the control experiment seen in figures 2(i-m) to assess whether reactivation of a tagged positive cocaine memory would affect locomotion or measures of anxiety and again, found no group differences. In our previous work (Liu et al., 2012) we have shown that artificial reactivation of a neutral memory also did not induce any behavioral changes. Indeed, we have shown that reactivation of a negative memory could induce (Liu et al., 2012) or maintain (Fig 2d) freezing. However, this would not be considered a confounding explanation for why a positive memory is able to interfere with a negative memory given that this is inferred from a decrease in freezing. With respect to the behavioral changes (e.g. freezing levels) that occur during the optical stimulation when we compare stimulation of a positive, neutral, or negative memory during the natural recall of a fear memory, we see that eYFP mice continue to freeze as expected, and that in the majority of

instances, activation of only a positively valenced memory results in real-time decreases in freezing. This seems to be a direct result of our manipulation; therefore, we do not see this as a confound, but as a difference emergent as a direct result of our manipulation. Additionally, in the current study, where we have shown that artificial stimulation of a tagged positive memory can interfere with the expression of conditioned fear during reconsolidation, we have made the important illustration that these effects are specific to the fear memory and do not affect other types of information e.g., a spatial reference memory.

6. Some data are a just a bit sketchy. For example, in Figure 1C, the Negative eYFP condition, there is a bimodal distribution such that 2 points are very high, and 3 points are very low. These animals are clearly very different, and in fact that difference is probably the largest numerical difference reported in the whole study. I would have redone this condition/experiment, but at the very least again, the authors should explain why we should trust data like this. Do these data even conform to the minimum requirements for the statistics?

We thank the reviewer for this observation. We agree there is a substantial amount of variability within this group. As a result, we have eliminated these two mice and added four additional mice to this group. We have replotted and reanalyzed the data and amended the stats in the associated figure caption. Also, the second paragraph of the results now reflects these changes. The results did change slightly, where there is now a significant main effect of valence driven primarily by the difference between the neutral-eYFP group and the negative-ChR2 group during reinstatement. This is also reflected by a significant valence x day interaction when we compare immediate shock to reinstatement, and each corresponding section in the manuscript has been updated accordingly.

7. What is the rationale for using only male mice?

We thank the reviewer for this question. We recognize that there is a significant male bias in neuroscience research by overrepresentation of the sole use of males in studies. We also recognize that the prevalence of PTSD is twice that in females compared to males. We originally tested our hypotheses in males because contextual fear conditioning in rodents is sexually dimorphic where males typically show higher freezing levels than females (Maren et al.; Wiltgen et al.; Yavas et al., 2021) and we wanted to make sure that our effect was large enough to warrant further experimentation. Notably, we also wanted to use female exposure as a positive stimulus and received comments that this might not necessarily be a positive experience for the females. While it is extremely unfortunate that we did not include females in this study, we are committed to being part of the changing tides on this issue and as a result, all of our current studies do include both males and females as we agree that it is absolutely essential for enhanced scientific discovery to do so.

8. Viruses in DG are known to kill adult born neurons. The authors should explain why this is not a problem in this study.

We thank the reviewer for this comment. This is definitely an important consideration. While recombinant adeno-associated viruses (AAV) have been widely used in neuroscience studies and have been shown to be safe and effective in human gene therapy it is true that there is evidence of AAV-induced cell death in immature dentate granule cells (Johnston et al. 2021). However, this study demonstrated that this was serotype specific. They found an effect that occurred within 48hrs post-injection. Following injection, our mice are left to recover for a minimum of 10 days. It is possible that prior to any behavioral component of the study, mice experience some cell death in adult-born neurons (ABNs). It has been shown that anesthetic agents also cause cell death (Strattmann et al., 2009). Fortunately, all of the mice within each experiment were given the same viral constructs (between experimental and controls) and the same

dose, which we highlight here was included in all our controls. Additionally, in our previous work (Doucette et al., 2020) we have shown that chronic optical stimulation of a tagged memory did not affect the number of doublecortin-positive cells which is a neural marker for ABNs. More specifically, chronically stimulating positive, neutral, or negative dorsal DG ensembles had no effect on neurogenesis and since we used the same AAVs in the current study we expected no adverse effects to occur. Moreover, in Ramirez et al. (2015) we observed that chronically activating a positive memory actually rescued stress-induced impairments on neurogenesis, and nonetheless appreciate the reviewer's poignant comments.

9. "Activation of randomly labelled dDG neurons is sufficient to disrupt fear consolidation."

"Reconsolidation-interference can be effectively achieved by activating ensembles that are not connected to an engram of a particular valence if enough cells are activated." The general reader may conclude that the experiments using memory insertion as disruption are meaningless, as any random messing around in the DG will yield the same effect, which is not unexpected and not very interesting. The authors need to do a better job of clarifying this for the reader.

We thank the reviewer for this comment and agree it is a valid concern. When we designed this study, we included this experiment with the hypothesis that it would not work and admittedly did not expect that it would be so effective. Nonetheless, we believe that there may be several factors worth considering here.

1) That valence does matter and that our reframing of the results in terms of the positive prediction error can help explain why the fear memory can be weakened through reactivation of a positive memory. 2) That stimulation protocols such as TMS, DBS, and ECT perturb the system in a way, which may provide a reset signal. That so long as the set of cells activated are sufficiently different than the fear memory engram cells and that there is a sufficient number of these cells activated (may even include those from positive emotional experiences), that this creates a difference in the expectation of the animal allowing for reconsolidation processes to occur thereby weakening the fear memory altering the associated engram. These novel modulation strategies involving circuit perturbations point to the dDG as a potential therapeutic node and may prove to be of high clinical significance for mechanistically guided psychiatric therapeutics. Moreover, we plan to use these strategies as well as develop other novel neuromodulatory strategies by targeting different brain regions and circuits. The fact that perturbation of the dDG cells can result in weakening of the fear memory implies that from a translational perspective, it may be possible in the future to use methods such as TMS or DBS to achieve similar results without the necessity of doing optogenetics in humans. We are very excited about these results and have included a paragraph in the discussion to summarize these thoughts so as to clarify for the reader what the take home message of the paper is.

10. If the authors can provide a bit more speculation about explanation of mechanism – why, for example, different valences of interfering memories have different effects – that could help make the discussion a little more satisfying than it is currently.

We thank the reviewer for this comment. We have thought about this a lot and believe that our results can be somewhat explained in terms of behavioral intervention and the expectations of the animal within the framework of negative and positive prediction errors and when the memory would be the most malleable. In the results section we have added a paragraph in the discussion. See below:

"Our results can be explained within the framework of negative and positive prediction errors possessing the ability to modulate the fear memory by strengthening or weakening it respectively. Prediction errors occur when there is a mismatch between what is expected and what happens. When an organism encounters something unexpected in their environment, this can drive memory-updating processes by triggering memory destabilization (Lee et al., 2018). This change in contingency promotes the

disengagement of established representations in favor of novel representations (Grella et al., 2019). During fear memory recall, mice are returned to the fear conditioning context in which they exhibit conditioned fear due to their expectation of being shocked as before. When they don't receive a shock, this contradicts their expectations thereby promoting reconsolidation. One theory is that providing them with a negative experience (e.g., artificial reactivation of a negative memory such as fear conditioning in a different context), likely resulted in their expectations being matched, and in no prediction error occurring thus did not lead to reconsolidation. Or, a negative prediction error may have occurred (actual was worse than expected) where the natural recall of the fear memory in context A plus the artificial reactivation of the fear memory from context C led to reconsolidation-based strengthening of the fear memory from context A. Contrastingly, when mice received artificial stimulation of a positive memory, this resulted in a positive prediction error (actual was better than expected). Essentially, when mice received artificial stimulation of a tagged memory, it fell along a continuum of valence where the more positive the experience, the greater the magnitude of the positive prediction error that ensued and the more likely this was to induce destabilization / memory-updating processes, which correlated with weakening the fear memory and decreases in fear expression."

11. The interfering-memory approach is suggested as a target for therapy for PTSD. Many readers will know that optogenetics is not feasible in humans. So how exactly will this work translate? Opto in the future? Drugs that target the DG? What's wrong with drugs that have been used previously, e.g., propranolol? Does this kind of approach bring any advantage above and beyond propranolol? More of this in the Discussion is necessary especially since the paper is set up to be all about PTSD.

We thank the reviewer for these comments. We agree, it is an important time in biomedical research and these questions are timely. A wonderful paper, recently published in *Cell* (Liston et al., 2022), entitled "Understanding the biological basis of psychiatric disease: What's next?" explores many of these questions - fittingly stating that "psychiatric disease is one of the greatest health challenges of our time". They go on to say that "The pipeline for conceptually novel therapeutics remains low, in part because uncovering the biological mechanisms of psychiatric disease has been difficult.". They describe a shift toward mechanism-guided psychiatric therapeutics and an urgent need for identifying new treatment targets. For instance, antidepressants developed 7 decades ago are still prescribed today, but are only effective in 33% of individuals with depression while depression continues to be a leading cause of disability. The field is poised for transformation, and this involves mechanistically informed treatment strategies driven by new technologies. We have now included this paper in the reference list and in the discussion.

Treatments such as propranolol take into account that norepinephrine (NE) may play a role in the formation of traumatic memories, and thus in the development of PTSD (Pitman et al., 1989). The idea is that NE enhances the formation of the negative associative memory i.e., strengthening conditioning (Cahill et al., 1994). Which is why propranolol, an FDA-approved beta blocker commonly used heart medication, has had so much attention for the possible treatment and prevention of PTSD. Because it inhibits NE by acting as a competitive antagonist on beta-adrenergic receptors (Srinivasan et al., 2019). While some studies have shown that propranolol can be effective in diminishing the emotionally enhancing effect of NE on memory, there have been mixed results and poor translation to humans. In the second paragraph of our introduction we state "Despite the long history of experimental reconsolidation-related interventions using a variety of pharmacological agents, behavioral treatments and stimulation protocols to disrupt or enhance memory^{6,7,8}, these studies have yielded mixed results." In the discussion we write "Currently, the majority of pharmacological and cognitive behavioral treatments used to treat disorders of emotional memory typically only affect the strength of the affective response while the original fear memory is left intact⁴⁸". Both these statements are meant to include propranolol. A recent

meta-analysis found that propranolol used as a preventative measure for PTSD following trauma, did not significantly reduce the risk for subsequent PTSD or acute stress disorder compared to placebo or no treatment. Moreover, we have shown in this study that our reconsolidation-based optical manipulation physically alters the original memory and studies involving propranolol suggest that the intensity of the memory is diminished but that the memory itself is not altered. Therefore, on this basis, our novel strategy may prove to be surprisingly effective.

While optogenetics may not be feasible in humans, (although it is being used in humans in some cases(e.g., it has entered clinical trials for conditions such as Retinitis Pigmentosa; White et al., 2020), our work is aimed at “the development and refinement of novel modulation strategies” which we expand on in the last paragraph of our introduction. For instance, circuit-based investigations such as these may open the doors for mechanistic insight into behavioral disorders providing new targets. The problem becomes not, what, but how do we do this in humans? And this work can inform current therapeutic interventions with respect to temporal parameters. As we see in this study, timing is important. With mice, we can artificially reactivate a memory, but with humans, we can potentially just ask them to remember, and it might be crucial to therapeutic outcomes “when” we ask them. Consequently, we have expanded our discussion to include more of this topic, as we agree with the reviewer, it places the findings in a larger context, and thank the reviewer for facilitating these edits.

More minor comments

12. Reference to data figures in the Results section should be made in sentences referring to the data, not to restatements of the methods.

We thank the reviewer for this comment. We have fixed this throughout the results.

13. Check rules for hyphenation.

We thank the reviewer for this comment. We have gone through the manuscript and removed many of the hyphens in phrases such as custom-built, off-DOX, eYFP-controls, ChR2-mice, and reward-location. We kept them for phrases like reconsolidated-based, self-stimulation, stress-induced, and adeno-associated.

Reviewer #3 (Remarks to the Author):

Summary

Grella et al., test the hypothesis that positively-valenced memories (exposure of a female conspecific, delivery of cocaine) formed in a novel environment can interfere with the reconsolidation of an aversive memory (e.g., a conditioned contextual fear memory). The authors approach this goal by combining the Tet-tag system, optogenetics, extensive behavioral testing, and immunohistochemistry in the dorsal dentate gyrus (dDG). The overarching question posed by this research – can rewarding experiences counteract negative experiences or even “re-write” them? – is both important and timely. The manuscript provides a plethora of interesting data. However, a number of important concerns make it difficult to support all of the conclusions drawn by the authors. These are highlighted below. Most warrant significant revisions to the text, not necessarily new experiments.

We thank the reviewer for these laudatory comments. We have done our best to address each of the reviewer’s concerns.

Concerns

1) The single most important issue is the question of whether or not the authors believe that the engram for a contextual fear memory is indeed stored in DG. There is an entire literature pointing to fear memory

engrams being stored in the amygdala, with engrams for spatial/contextual information being stored in the hippocampus (e.g., starting with Kim and Fanselow, 1992, to Rashid et al., 2016, and numerous more). In other words, one common view is that the hippocampus encodes contextual information, which then gets sent to the amygdala for formation of a contextual fear memory engram. These data (as well as previous data published by this research group), seem to contradict this. While this reviewer is not suggesting more experiments to address this discrepancy, I believe that a section on this matter in the discussion is highly warranted.

We thank the reviewer for this comment. We agree that the amygdala is an integral part of both cued and contextual fear-conditioning. We have included a sentence in the discussion that ties our findings in with the previous literature and states the significance of the basolateral amygdala (BLA) and how we plan to explore our manipulation in this brain region in future studies.

To address why we specifically targeted the DG, we have included a clearer explanation of our rationale for this brain region in the intro and added the appropriate citations. We have included the word “fear” after “contextual” and before “memories” to make this distinction clearer (middle of 3rd paragraph in the intro) and added additional references.

Previous studies (Saxe et al., 2006; Khierbek et al., 2013) including two more which we have added to the text (Bernier et al., 2017; Ressler et al., 2021), suggest that the hippocampus, specifically the DG, plays a role in encoding contextual fear memories, which may even involve neurogenesis (Seo et al., 2015). Moreover, several studies suggest that the DG and BLA work in concert and it is likely that both regions are simultaneously important for fear learning as engrams have been found to be widely distributed throughout the brain (Roy et al., 2022). Yavas et al., (2021) suggests that contextual fear-conditioning is thought to involve the integration of multiple cues that encompass the context, which must be integrated into a coherent representation. A process that requires the hippocampus but where this representation is then communicated to the BLA where it can be associated with shock. Moreover, our previous research (Redondo et al., 2014; Ramirez et al., 2015) also demonstrated that the pathway from the DG to the BLA is necessary, not only for contextual fear conditioning but also to acutely rescue stress-induced, depression-related behaviors. In Ramirez et al., (2015), this was specifically achieved by optogenetically reactivating DG cells that were previously active during a positive experience where downstream (BLA and NAcc shell neurons were found to also be active. Therefore, the current work was conducted as a follow up to the Ramirez et al., (2015) paper based on the memory modulation strategies used, and as a follow up to our previous findings showing that reactivation of a fear memory tagged in the dorsal DG is sufficient to induce freezing (Liu et al., 2012). For this reason, we specifically chose to target the dorsal DG for our manipulation.

Additionally, we want to promote the idea that valence (e.g., negative memories) can be considered an aspect of “context” and that contextual information can include much more than simply spatial information. For example, with respect to PTSD specifically, there is a proposed impairment in processing contextual cues signifying safety (cues associated with the absence of threat) (likely involving the DG-PFC connections), as well as a hyper-responsiveness to trauma-related stimuli. These associations are formed within the hippocampus (Sripada et al., 2013) and more specifically disambiguated within the DG (Liberzon & Abelson, 2016; Lissemore et al., 2020). Therefore, fear generalization likely involves an impairment within the DG (constituting part of the greater DG-BLA-mpFC circuitry). In PTSD patients in particular, context may have a reduced capacity to modulate fear and safety and this may be reflected in neuroanatomical differences such as reduced hippocampal volume (Sripada et al., 2013). Moreover, extinction is highly dependent on the hippocampus (Lacagnina et al., 2019) and PTSD patients exhibit impairments in extinction learning (Milad et al., 2008).

2) The authors link their results to relevance for PTSD, indeed they state in the intro that “pathological conditioned fear can occur for decades even in the absence of the exact context in which the traumatic event took place.” Given that they are looking at contextual fear in particular here, it seems that their findings are more relevant to specific phobias rather than PTSD. Related to this, in the second paragraph of the intro, the authors link their findings to “maladaptive fear”, however, again, they are not modeling a maladaptive fear here, but are rather examining an adaptive, conditioned context fear. Text should be adjusted to reflect this.

We thank the reviewer for this excellent question and their insight. Convergent evidence suggests that PTSD is associated with various abnormalities in fear-associated learning, including greater acquisition of conditioned fear combined with exaggerated fear responses. However, this can also be extended to include overgeneralization of conditioning, enhanced stress reactivity, impaired inhibitory learning, and impaired extinction (Milad et al., 2008, 2009; Jovanovic et al., 2009; Sripada et al., 2013). We have included a sentence in the first paragraph of the intro to explain that contextual fear conditioning is often used to model aspects of PTSD such as exaggerated fear / heightened stress reactivity and generalization - behaviors which constitute maladaptive conditioning and added the relevant citations.

One hallmark feature of PTSD is rumination and the ability for traumatic memories to be intrusive. This work is aimed at trying to modulate negative memories such as those, hence its relevance to PTSD. We have included a sentence in the discussion to reflect this. We have also amended our intro to include phobias under the umbrella of anxiety disorders relevant to this work. Importantly, we have also attempted to make it clearer in the results and not just the methods that we used: an immediate shock delivered in a different context to reinstate fear conditioned responses (Fanselow 1986, 1990; Fanselow et al., 1994). This methodology allowed us to model fear generalization and heightened stress reactivity as an example of maladaptive conditioning given that fear responses were extinguished in context A, and mice demonstrated fear responses in this context following a stressor given in a different context.

3) Supp figure 1 is a lot of data points with no information as to which data point belongs to which animal in which group. There are a few data points showing pre-conditioning freezing (!). Which animals were these? Please re-graph this to include group identities.

We thank the reviewer for this comment and for noticing those data points where there was high pre-shock freezing - these animals were in fact the mice that were fear conditioned in a different context earlier (so received FC twice) (Fig 2 negative groups) so they should not have been included. However, instead of removing those data points we took out the figure entirely. We agree with the reviewer in that the information would be more useful if it was divided and presented by group. We have remade this figure (New Supplemental Figure 1) to show fear acquisition curves with new analyses to illustrate the stepwise manner of freezing elicited by using the 4-protocol. The manner in which the data is expressed allows the reader to easily see that most mice follow this same pattern irrespective of the experiment. Interestingly though, the one exception is for the mice in Fig 5 where we trained them on a spatial reference memory. We believe that they demonstrated less freezing during fear-conditioning (since it was across all groups) by virtue of the fact that they were handled so much more due to being trained for weeks on the spatial task. We have included this observation in the results section of this experiment. (Fig 5).

4) The authors spend a lot of time highlighting reconsolidation. They present data on reinstatement and spontaneous recovery as indirect read outs of whether or not reconsolidation has been interrupted. However, their extinction sessions (in particular the first one), are also powerful read outs. But because

this data is being collapsed across freezing during the entire extinction session, the authors may be missing out on important data. The authors should instead break down the extinction sessions so that freezing during the first few minutes (and the effects of their manipulations on these time points) can be investigated. This is especially important as the authors repeatedly make claims about extinction “rate” (e.g. Fig 1j), but never present any data or statistical analyses looking at/comparing the actual rate (i.e. extinction curve). This data can be put in the supplemental.

We thank the reviewer for this suggestion. We have reorganized the data and included a new Supplemental Figure 1, which shows the fear acquisition curves and the extinction data for the first 3 minutes of the first context exposure and the last 3 min of extinction training on day 2. This was done for all the experiments. Statistical analyses are presented in the figure caption and in the results section where applicable. Additionally, we ran several new control experiments that were included in the other supplemental figures - for these figures we have included both the acquisition curves and the extinction data showing the first and last 3 min as well. These results and stats have been added to the results section and in the figure captions respectively. Generally speaking, memory modulation observed by looking at the first 3 min of extinction was stronger when we activated a random group of DG neurons with the CAMKIIa promoter virus rather than using the c-Fos-ttA virus to tag differently valenced experiences. However, when we look at positive memory reactivation in comparison to the No Light control groups (New Supplemental Figure 3) or simply on it own in comparison to eYFP control groups at either the first 10 min, the last 10 min or the middle 10 min (New Supplemental Figure 2), we do see some significant effects. So, in some cases it depends on the statistical test used to see the difference but overall, memory modulation appears strong enough to see a group difference at this important time point.

More specifically, we found that for all experiments there was a main effect of time. For experiment 1 (Figure 1, New Supplemental Fig 1a-c), there was a significant main effect of virus for mice that were stimulated in either the first or last half of the recall session, but post-hocs revealed no significant group differences during the first 3 min of extinction. For the mice in the spontaneous recovery experiment, we saw no main effect of virus, only time. For experiment 2 (figure 2, New Supplemental Fig 1d), there was a significant virus x time interaction but again no group differences in the first 3 min of extinction. For the VTA experiment (Figure 3, New Supplemental Fig 1e), there was significant main effect of VTA virus and a significant Time x DG Virus interaction and a significant group difference between the VTA Chr2 and VTA-eYFP groups in the last 3 min of extinction and between the VTA-CHR2-dDG-ChR2 group and the VTA-CHR2-dDG-eYFP group but no group differences in the first 3 min of extinction. We began to see differences in the first 3 min of extinction when we looked at stimulating the non-engram DG cells with the CAMKIIa promoter virus (Figure 4). There was again a significant effect of time, but also virus and a significant difference between the diluted Chr2 group and diluted eYFP group (New Supplemental Fig 1f). For the maze experiment (Figure 5), where we used this same viral strategy, there was a significant time x virus interaction with general group differences between the Chr2 and eYFP groups despite reminder status (New Supplemental Fig 1g). And for experiment 6 (Figure 6, New Supplemental Fig 1h) we see a main effect of time and a distinct group difference at both the first and last 3 minutes of extinction. When we compare the Chr2 and eYFP groups that received stimulation in the first half of the recall session (Figure 1) with the no light groups (New Supplemental Fig 3), we do then see a significant Light x Virus interaction with a group difference observed. When we compared time of stimulation: either first, middle, or last half of the session (New Supplemental Fig 2) we found a significant time x virus interaction with a group difference only in the group that got stimulation in the first 10 min.

5) Fear acquisition curves (instead of bar graphs) should be shown. This will allow the authors to determine whether any of their manipulations affect the rate of fear acquisition.

We thank the reviewer for this comment. We conducted the fear conditioning component of all our experiments prior to any of our manipulations. However, we agree that this would provide more detailed information and allow us to examine these curves across experiments. Therefore, we have included a new New Supplemental Figure Fig 1 that shows the fear acquisition curves for all the experiments we ran and also included them in the other supplemental figures for the control experiments we added. These results and analyses have been added to the results section where application and the associated figure captions. We are especially appreciative of the suggestion to reorganize the data this way, since in replotting / analyzing the data, it allowed us to notice an interesting effect that we would have not seen otherwise. In Fig 5 where we trained mice on the spatial reference task, these mice demonstrated less freezing during fear-conditioning (across all groups), which we believe to be a result of the extra handling they received having been trained for weeks on this task. We have included this interesting observation in the results section of that experiment. (Fig 5).

6) The first experiment reveals that there are interfering effects with “neutral” experiences. The authors then shift to a homecage exposure in experiment 2, claiming that the original “neutral experience” in a novel clean cage may actually be positive. If anything, exposure to a novel environment might produce some neophobia, however the authors do not address this.

We thank the reviewer for this comment and appreciate their insight. We hope the changes we have implemented successfully incorporate the necessary nuance of this topic. Initially, the main reason we suggested that this experience may be rewarding rather than aversive was due to the fact that during exploration of a novel environment, the brain's reward systems are activated (Krebs et al., 2011; Parkitna et al., 2013; Costa et al., 2014; Lee et al., 2021), and mice have a natural proclivity to engage in the exploration of novel objects and environments with a lack of this behavior used often as a measure of anhedonia (Bevins and Besheer, 2005). However, it is true that dopaminergic systems are also activated during the processing of aversive stimuli (Vander Weele et al., 2018) and simply during any experience that is perceived as salient (Kutlu et al., 2021). We appreciate that the relationship between approach to and avoidance of novelty arising from neophobia is indeed complex. This is the same relationship that makes new dams afraid of their litter but also what allows them to provide care for them. The main point that we wanted to get across was that the experience potentially carried a valence that was not neutral. We agree, especially after reanalyzing the results in Figs 1c, & 1f, which showed a significant difference between the negative and neutral experiences in terms of freezing. To reflect this important distinction, we have amended the text, specifically in the “Valence Matters” section of the results stating that we “first we tested if the novel clean cage experience was indeed “neutral,” rather than positive or negative given that novel stimuli can engage a complicated set of approach and avoidance dopaminergic pathways related to salience, reward, and neophobia.” We have also added additional references in this section.

7) Related to point 6. The “homecage “neutral” environment represents an environment that the animal has been in for an extended period of time. Such a condition is often used as a homecage control wherein almost no cfos expression is present in the dorsal hippocampus (e.g., work from Barnes lab). How are the authors “tagging” activated neurons in this condition if there are usually almost no cfos+ cells in this homecage condition??

We thank the reviewers for this comment. Other studies using cellular compartmental analysis of temporal *in situ* hybridization (Marrone et al., 2008) have shown low immediate early gene (IEG) mRNA expression during home cage experiences, where, to induce IEG expression, the animals need to specifically traverse an environment (Grella et al., 2019). In our previous work (Liu et al, 2012), using a c-Fos-tTA transgenic mouse rather than virally transfecting cells, we also found that a home cage experience

resulted in approximately 2% of cells being tagged. However, in the current study using an all virus-based design, we found that approximately 8% of DAPI-labeled neurons in the dorsal DG were tagged as engram cells ($M=8.015$, $SD=0.572$) (Figure 7). We included this information in the second paragraph of the section titled “Rewriting the original fear memory” in the results. We did our best to keep these mice undisturbed during the tagging of the home cage experience, where signage was placed on the cages for the vet staff not to disturb the mice or lift the cage tops. The difference in engram size we see in the current study compared to these other studies may reflect something environmental or may be due to the difference in methodology used with our viral system. Nonetheless, since there were no differences in the engram size between the homecage experience and the other experiences, this allowed us to reactivate these cells successfully.

8) It is interesting that memory “interference” is only effective when the authors activate positively-valenced, tagged ensembles at the beginning of the reactivation test. However, their explanation that this is because the second half of the reactivation session ends with transport back to the homecage remains untested. In order to test this, the authors should run a control experiment where activation occurs in the middle of the 20-minute reactivation session (eg min 5-15), and examine whether interference is still intact. Otherwise, they cannot rule out that the reason the first 10 min activation works better is because animals haven’t begun to extinguish their fear.

We thank the reviewer for this comment and as a result have restructured the entire first paragraph of the results section to make it explicitly clear why we chose the time intervals we did, which we hope improves the readability of the paper. We used a 20 min recall test so that our manipulation would occur during the reconsolidation period of the fear memory rather than a longer session where extinction learning might begin to take place (Suzuki et al., 2004). We simultaneously stimulated the tagged dDG ensembles during only the first or last half of the session rather than the entire session, as we did not want to risk damage to the brain via heat (Arias-Gil et al., 2016) and wanted to compare light on and light off periods. We wanted to compare the manipulation when it occurred post-memory reactivation (last 10 min) and when it did not (first 10 min). We initially hypothesized that reactivating a positive memory during the last half of the recall session would promote reconsolidation because the fear memory would be online, as opposed to starting the stimulation simultaneously with memory reactivation. We chose a 10 min interval of optical stimulation based on previous experiments where we saw promising results (Ramirez et al., 2015; Chen et al., Doucette et al.) where the same interval was used. However, the one challenge it presented, as mentioned, was that the animals in the last 10-min group were placed into their home cages directly after the session. As mentioned, we did not want to keep them in the context for longer as we did not want extinction learning to occur during this session.

To address this, we have included the additional experiment / New Supplemental Figure 2 suggested by the reviewer where mice were given stimulation during the middle portion of the session. They did not receive stimulation for the first 5 min and then for 10 consecutive minutes they were given optical reactivation of a positive memory (ChR2 and eYFP mice), and were then left in the context for an additional 5 minutes without stimulation. This clever design ensures the session length, and the stimulation length are kept constant. And yet allows us to see pre-stimulation freezing and post-stimulation freezing. It is powerful with the comparison to the other two groups so that we can still look at the effect of our manipulation when it occurs simultaneously with natural fear-memory reactivation, and post-reactivation. We found that this method was more effective than stimulating the mice in the latter half of the session but not quite as good as when the stimulation occurred at the start of the session. We believe the most parsimonious explanation for this is that mice were shocked between minute 3-7 during the fear-conditioning session and that the fear memory may be most malleable during the recall session around that same time, as this is when the highest level of shock expectation would theoretically occur. In

the third paragraph we discuss how prediction errors drive reconsolidation and how artificial reactivation of a negative memory likely does not result in a prediction error or possibly results in a negative prediction error leading to strengthening of the fear memory while artificial reactivation of a positive memory likely leads to a positive prediction error resulting in reconsolidation and a weakening of the fear memory. Essentially, we have described the artificially tagged memory as falling along a continuum of valence where the more positive the experience, the greater the magnitude of the positive prediction error that ensues and that negative and positive prediction errors can modulate the fear memory in such a way that it strengthens or weakens it respectively. As we expand on in this discussion, we posit that prediction errors modulate reconsolidation, and thus it makes sense that the animal's expectations would really drive the reconsolidation process in terms of timing as well. We have amended the methods section on Recall to include this experiment. We have also added these results to the results section, included the stats in the new figure caption, and have incorporated the implications of the experiment into the discussion. .

9) In the discussion, the authors bring up the interesting possibility that there may be less interference of a contextual fear memory by an aversive experience because there is more overlap of engrams. This would be consistent with the data that the authors show regarding activation of a random subset of DG neurons and the ability of this activation to similarly interfere with contextual fear memory reconsolidation. Hence, this is the most parsimonious explanation for the data. For this reason, the real finding of the paper is that fear memory reconsolidation can be perturbed if non-overlapping cell populations are activated in DG during retrieval of that fear memory, rather than anything specific about a positively-valenced experience.

We thank the reviewer for this comment. To recap: in the second paragraph of the "Rewriting the original fear memory" section of the results we stated that "We observed a greater percentage of overlap for negative experiences compared to neutral experiences suggesting the negative experiences shared higher similarity with the fear memory compared to the neutral experiences." referring to Figure 7k. Then in the second paragraph of the discussion we stated "Of note, it is possible the engrams associated with negative experiences, especially fear conditioning, are highly similar to the fear memory thereby providing less interference. Indeed, we observed that dDG cells processing negative memories overlapped more with the fear memory at reinstatement in comparison to the neutral memories."

We agree with the reviewer in that it is interesting to see that the similarity between fear engrams may be what's preventing reactivation of those engrams from interfering with the fear memory, and how this is consistent with the ability of the activation of random DG neurons to interfere with the fear memory. However, one thing that we must note, and the main reason why we did not suggest that this was likely the mechanism by which our manipulation was occurring, is that the difference in overlaps we see is between negatively valenced engrams and neutral engrams but not between negatively valenced engrams and positively valenced engrams. This suggests that the mechanism by which positively valenced engrams are able to interfere with the fear memory during reconsolidation is not due to the fact that they are significantly non-overlapping with the fear memory. Statistically, overlap is similar between negative and positive engrams, and between neutral and positive engrams, but not between negative and neutral engrams. To clear up any possible confusion, in the results section we have added a section that says "We observed a greater percentage of overlap for negative experiences compared to neutral experiences suggesting the negative experiences shared higher similarity with the fear memory compared to the neutral experiences. However, no differences were observed between the percentage of overlap for negative experiences compared to positive experiences, or positive experiences compared to neutral experiences. So, while we can infer that the degree of overlap between negative experiences and the fear memory may be what's preventing reactivation of those engrams from interfering with the fear

memory, we cannot assume that our ability to successfully weaken a fear memory by reactivating a positive memory derives solely from a disparity in these populations.” We have also elaborated our discussion on this topic and have included an alternate explanation involving positive prediction error as the most likely reason for why positive engrams are able to interfere with a fear memory despite any level of similarity to the original fear engram.

Minor issues

1) In the intro (first paragraph) the authors state that footshocks elicit freezing, they don't (they elicit activity bursts). Only the CS (in this case a context) elicit freezing (a CR) via conditioning.

We thank the reviewer for this comment and have changed the wording in this paragraph to correct this statement.

2) Methods. Were mice run during the light or dark part of the cycle?

We thank the reviewer for this question. We have amended the Animals section of the methods to include this information.

General changes that have been made

- Animal numbers were amended to include the new control experiments
- All of the IS-RE comparisons for every figure have been redone to show post-hoc comparisons on the graph.
- Acknowledgements and contributions section has been updated
- General editing

REVIEWERS' COMMENTS

Reviewer #1 (Remarks to the Author):

The authors have made an exemplary effort revising the manuscript. I have no further concerns. This body of work they have produced will make a significant impact on the reconsolidation field and I believe it should be published.

Reviewer #2 (Remarks to the Author):

The authors have responded well to my comments and I am happy for the manuscript to go forward as it is now.

Reviewer #3 (Remarks to the Author):

For the most part, the authors adequately addressed all of my concerns, and I am looking forward to seeing this work published!

The only lingering issue is that in the response to Concern #1, the authors replied (second paragraph) with a very interesting comment about contexts being partially comprised by their valence (pos vs. neg). However, when I looked through the revised text, this was not mentioned (it is only in the response to reviewers). I think the MS would really benefit from a brief discussion regarding this notion, as it helps to address the issue of what exactly is being encoded by DG.

REVIEWERS' COMMENTS

Reviewer #1 (Remarks to the Author):

The authors have made an exemplary effort revising the manuscript. I have no further concerns. This body of work they have produced will make a significant impact on the reconsolidation field and I believe it should be published.

We thank the reviewer for taking the time to review our paper.

Reviewer #2 (Remarks to the Author):

The authors have responded well to my comments and I am happy for the manuscript to go forward as it is now.

We thank the reviewer for taking the time to review our paper.

Reviewer #3 (Remarks to the Author):

For the most part, the authors adequately addressed all of my concerns, and I am looking forward to seeing this work published!

The only lingering issue is that in the response to Concern #1, the authors replied (second paragraph) with a very interesting comment about contexts being partially comprised by their valence (pos vs. neg). However, when I looked through the revised text, this was not mentioned (it is only in the response to reviewers). I think the MS would really benefit from a brief discussion regarding this notion, as it helps to address the issue of what exactly is being encoded by DG.

We thank the reviewer for taking the time to review our paper.

We have added a paragraph in the introduction addressing the specific contribution of the DG to contextual processing and how valence fits into that.

Additional changes made:

We have made some other changes throughout the paper to comply with the formatting of the journal. Namely:

- The tense in the abstract has been changed, abbreviations have been removed, and the species we used was included
- We have added an additional sentence at the end of the introduction summarizing our main findings
- Sections of the paper have been reorganized to match the formatting guidelines
- We have added detailed information in the methods such as software, animal conditions, statistical information, and a data availability section that includes a link to the custom Matlab code used in our experiments and provided the raw data source file
- We have amended our Biorender statement in the acknowledgements section to include the exact content and figures Biorender was used for
- For Figure 7, we added missing n values to the figure legend, included the individual data points, and for Supplementary Figure 2, we noticed that two panels needed to be switched.
- We edited the figure legends to reduce the word count as per the formatting guidelines
- We edited the references and also added DOIs
- We separated the Supplementary info from the main manuscript